# Downregulating carnitine palmitoyl transferase 1 affects disease progression in the SOD1 G93A mouse model of ALS

Michael Sloth Trabjerg [1], Dennis Christian Andersen[1], Pam Huntjens[1], Kirsten Egelund Oklinski[1], Luise Bolther[1], Jonas Laugård Hald[1], Amalie Elton Baisgaard[1], Kasper Mørk[1], Nikolaj Warming[1], Ulla Bismark Kullab[1], Lona John Kroese[2], Colin Eliot Jason Pritchard[2], Ivo Johan Huijbers [3] & John Dirk Vestergaard Nieland [1✉]

Amyotrophic lateral sclerosis (ALS) is a fatal motor neuron disease characterized by death of motor neurons. The etiology and pathogenesis remains elusive despite decades of intensive research. Herein, we report that dysregulated metabolism plays a central role in the SOD1 G93A mouse model mimicking ALS. Specifically, we report that the activity of carnitine palmitoyl transferase 1 (CPT1) lipid metabolism is associated with disease progression. Downregulation of CPT1 activity by pharmacological and genetic methods results in amelioration of disease symptoms, inflammation, oxidative stress and mitochondrial function, whereas upregulation by high-fat diet or corticosterone results in a more aggressive disease progression. Finally, we show that downregulating CPT1 shifts the gut microbiota communities towards a protective phenotype in SOD1 G93A mice. These findings reveal that metabolism, and specifically CPT1 lipid metabolism plays a central role in the SOD1 G93A mouse model and shows that CPT1 might be a therapeutic target in ALS.

[1] Department of Health Science and Technology, Aalborg University, Aalborg, Denmark. [2] Mouse Clinic for Cancer and Aging Research, Transgenic Facility, The Netherlands Cancer Institute, Amsterdam, The Netherlands. [3] Swammerdam Institute for Life Sciences, University of Amsterdam, Amsterdam, The Netherlands. ✉email: Jdn@hst.aau.dk

Amyotrophic lateral sclerosis (ALS) is a progressive neuro-degenerative disease affecting both the central nervous system (CNS) and periphery[1]. It is characterized by degeneration of upper- and lower-motor neurons resulting in death caused by respiratory failure within 3 years in many cases[1]. The incidence of ALS is 1–2 cases per year per 100,000 people with a cumulative lifetime risk of 1:400[2]. ALS exists in a sporadic- and a familial form, which accounts for 10% of all ALS cases[1]. The male to female ratio is ~2:1 for the sporadic form and 1:1 in the familiar form[1]. The first mutation to be associated with the familiar form of ALS was in the SOD1 gene coding for the protein superoxide dismutase 1 (SOD1)[3]. Mutations in SOD1 are responsible for ~20% of all familial cases and 1–3% of all sporadic cases[1]. The most prevalent mutation in SOD1 is the glycine to alanine conversion at the 93rd codon resulting in a toxic gain-of-function[4]. However, in the last decade, several genes have been associated with a risk of ALS, and the disease has a broad heterogeneity with several clinical phenotypes[5]. The etiology of ALS remains elusive, but mechanisms causing the motor neuron degeneration in ALS include deposition of protein- and RNA aggregates, oxidative stress, endoplasmic reticulum stress, glutamate excitotoxicity, mitochondrial dysfunction, and neuroinflammation[1,6]. At the moment, there is only two FDA approved drugs, both of them with little- to no therapeutic effect on survival of the patients[5,7]. Interestingly, enhanced lipid metabolism and decreased glucose metabolism has been observed in ALS patients and animal models and have been associated with disease progression[8]. Dupuis and colleagues[9] investigated whether energy homeostasis is defective in two transgenic ALS lines, the SOD1 G86R and SOD1 G93A mice, and found several indications of dysregulated metabolism. At day 105, the lipid concentration was significant lower in the transgenic SOD1 G93A mice compared to healthy control animals. Dodge et al.[10] examined whether SOD1 G93A mice undergo metabolic changes during disease. They showed that transgenic SOD1 G93A mice present several changes indicative of altered lipid metabolism, such as reduced circulating triglyceride levels and reduced fat mass. One of the key molecules in lipid metabolism is carnitine palmitoyl transferase 1 (CPT1), which facilitates the transport of medium-long chained fatty acids across the outer mitochondrial membrane to undergo $\beta$-oxidation to produce acetyl-coA. CPT1 exists in three isoforms; CPT1A, which is located in most of the body, CPT1B, which is solely located in the muscles, and adipose tissue and CPT1C, which is only present in the CNS but is associated with the endoplasmic reticulum[11]. Upregulated lipid metabolism is essential for the survival of immune cells, and it enhances multiple pathogenic processes, including inflammation, oxidative stress, and mitochondrial dysfunction[12–14]. Furthermore, production of acyl-CoA through upregulated $\beta$-oxidation results in a negative feedback to pyruvate, which results in decreased glucose metabolism, leading to a vicious cycle disrupting the homeostasis[15]. CPT1A and CPT1B have been found to be upregulated in the SOD1 mouse model[16–19]. Further, human populations, which have a significant reduction in CPT1A activity, have been found to have a decreased prevalence of CNS diseases such as multiple sclerosis and ALS[20,21]. We have previously shown that blocking of CPT1 by an irreversible CPT1 antagonist or by genetic inhibition show clinical-relevant effects in in vivo models of neurodegenerative diseases, diminishes inflammation, demyelination oxidative stress and increases mitochondrial biogenesis[20,22–24]. Further, Timmers et al.[25] have shown that a CPT1 antagonist upregulates glucose metabolism and insulin sensitivity. Recently it was shown that tissue-specific knock-out of Cpt1b in mouse muscles results in increased oxidative capacity, glucose metabolism, and insulin sensitivity[26].

Based on reported findings of the dysregulated metabolism in ALS patients and in vivo models of ALS, we tested the hypothesis that blocking or downregulation of lipid metabolism through CPT1 modulation affected disease progression in the SOD1 G93A mouse model of ALS. Herein, we report that pharmaceutical and genetic inhibition of CPT1 is able to delay disease progression in the SOD1 G93A model. Further, we report that environmental dysregulation of metabolism through saturated animal-based high-fat diet (HFD) and corticosterone results in a more aggressive disease progression. We also report that modulation of CPT1 results in alternations of the serum levels of glucose and lipids, inflammatory, oxidative stress, neuronal, glial and metabolic markers in the spinal cord and inflammatory, oxidative stress, metabolic and denervation markers in the tibialis anterior muscles. Finally, we report that modulation of CPT1 activity results in changes in the fecal gut microbiota, which is consistent with the increased recognition of the possible role of dysregulated microbiota in ALS and other neurodegenerative diseases. Altogether, these results suggest that the upregulated CPT1 lipid metabolism plays a critical role in the pathogenesis of the SOD1 G93A familiar form of ALS and that downregulation of CPT1 might be a potential target to restore the hyperactive metabolism.

## Results

**CPT1 antagonist delays disease onset and alleviate clinical symptoms in SOD1 G93A mice.** The SOD1 G93A (over-expressing human SOD1 G93A) mouse display progressive deficits in fine and gross motor function, as well as some degree of cognitive impairment[4,27]. The SOD1 G93A gene mutation was the first gene linked to ALS, providing an epidemiological foundation for the SOD1 G93A mouse model[3]. Defects in motor coordination become evident from the age of 70 days[27]. Several studies have indicated that the metabolism is dysregulated in the periphery and CNS in the SOD1 G93A model from early disease stages and onward[9,16,19,28]. Based on this, we tested the effect of blocking CPT1, which previously has been shown to shift the metabolism from lipid- to glucose metabolism and show clinical-relevant effects in a pilot study[24,25].

Female SOD1 G93A mice (SOD1) and their female wildtype littermates (Wt) were randomized into treatment with a CPT1 antagonist (etomoxir) (SOD1 + E, Wt+E) or placebo (SOD1 + P, Wt+P) from day 70 (Fig. 1a). Etomoxir targets both CPT1A and CPT1B by antagonistic mechanism in the periphery and CNS and is indicated to cross the BBB based on functional analysis and chemical structure. Motor function was evaluated weekly from day 70 by rotarod[29], hangwire[30], grip strength[24] test and neurological score[31], as previously validated in this model. Further, visuospatial memory and spontaneous activity were evaluated using the y-maze test- and cylinder test. The y-maze test has recently been validated in the SOD1 G93A model[27]. The spontaneous activity cylinder test has not been conducted in the SOD1 G93A model previously according to the authors' knowledge. The test has, however, been validated in several other mouse models as a measure of sensorimotor function- and spontaneous activity[32]. SOD1 G93A mice develop tremor in their hind legs as one of their characteristics[30]. Interestingly, the SOD1 + P mice (median onset = 89 days) developed tremor significantly earlier ($p$ = <0.0001) compared to the SOD1 + E mice (median onset = 106.5 days), indicating that etomoxir delayed disease onset (Fig. 1b). This was consistent with significantly lower neurological score from day 91 and throughout the experiment for the SOD1 + E mice (Fig. 1c). The rotarod test did not find any significant differences between the SOD1 groups. However, the SOD1 + P had a significantly lower latency to fall compared to the Wt group throughout the experiment, which was not the case for the SOD1 + E group (Fig. 1d). Interestingly, the SOD1 + E mice had a significant increase in their normalized grip strength following treatment with the irreversible CPT1 antagonist and

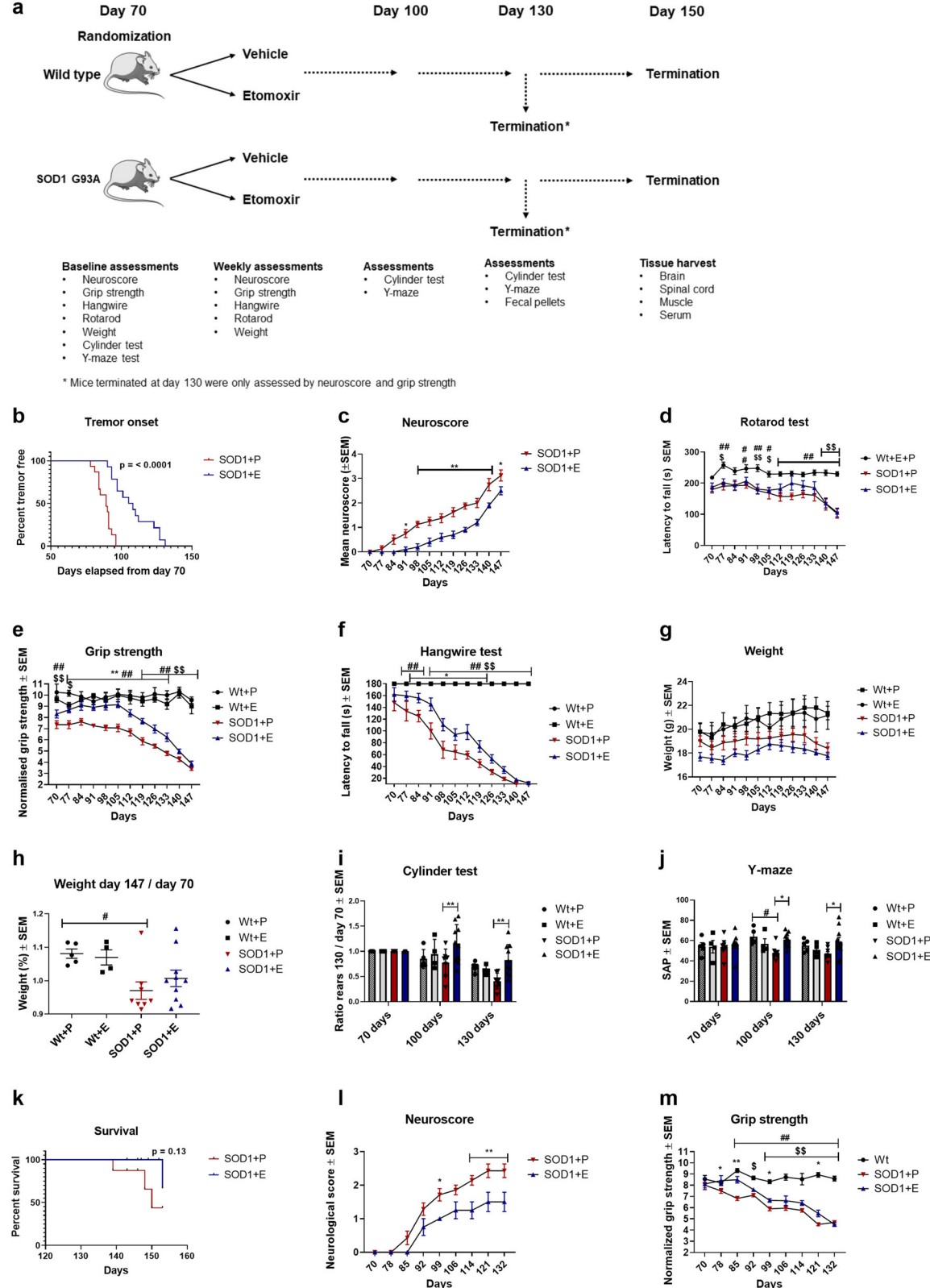

had a significant higher grip strength compared to SOD1 + P mice for 9 weeks (Fig. 1e). SOD1 + E mice had a significant higher latency to fall on the hangwire test for several weeks (Fig. 1f). SOD1 G93A mice lose weight during their disease progression, which was also the case in this study (Fig. 1g). SOD1 + P had a significant weight loss compared to their wildtype littermates, which was calculated as the weight ratio day

147/day 70 (Fig. 1h). The cylinder test showed that SOD1 + E mice had a significant higher number of rears at day 100 and day 130 compared to SOD1 + P mice (Fig. 1i) indicating a better sensorimotor function- and higher activity. SOD1 + P mice had a significantly lower spontaneous alternation percentage (SAP) in the y-maze at day 100 compared to Wt+P (Fig. 1j), indicating a loss of visuospatial memory. Strikingly, the SOD1 + E mice had a

**Fig. 1 Downregulating CPT1 by etomoxir ameliorate clinical symptoms and delays disease progression in the SOD1 G93A mouse model. a** Experimental setup, mice were randomized into treatment with etomoxir or placebo at approximately day 70 (baseline) and tested weekly until day. **b** Time point for onset of disease defined as visible tremor in hindlimbs ($n = 11$–15), log-rank survival analysis. **c** Neuroscore was assessed once a week. Mean neuroscore ± SEM ($n = 8$–10). **d** Rotarod test was performed once a week. Mean latency to fall ± SEM ($n = 4$–7). **e** Grip strength was measured once a week from day 70. Data are presented as mean normalized strength ± SEM ($n = 4$–10). **f** Hangwire test was performed once a week. Mean latency to fall of the grid ± SEM ($n = 4$–10). **g** Mean body weight expressed in grams ± SEM ($n = 4$–10). **h** Mean body weight ratio at day 147 compared to baseline ± SEM ($n = 4$–10), one-way ANOVA with Tukey post hoc test. **i** Cylinder test was performed at day 70, 100, and 130. Mean ratio rears between day 100, 130, and baseline ± SEM ($n = 4$–10). **j** Y-maze test was conducted at day 70, 100, and 130. Mean spontaneous alternation percentage ± SEM ($n = 4$–10). **k** Survival was defined as neuroscore below 4, ($n = 8$–10), log-rank survival analysis. If nothing else is noted, data was analyzed using repeated measure two-way ANOVA followed by Tukey post hoc test. **l** Mean neuroscore for SOD1 mice treated from day 70 and until day 130 ± SEM ($n = 4$–7). **m** Mean normalized grip strength for SOD1 mice treated from day 70 and until day 130 ± SEM ($n = 4$–7). Data are representative of three experiments. Repeated measure two-way ANOVA with Tukey post hoc test was performed if nothing else is stated. *Significant differences between SOD1 + E and SOD1 + P, #significant differences between SOD1 G93A + P and Wt, $significant differences between SOD1 + E and Wt. *$p \leq 0.05$; **$p \leq 0.01$; #$p \leq 0.05$; ##$p \leq 0.01$; $$p \leq 0.05$; $$$p \leq 0.01$ WT = wild-type, SOD1 = SOD1 G93A genotype, E = etomoxir, P = placebo. Mouse figure used in experimental setup figures were obtained from; https://smart.servier.com/ and used according to the Creative Commons Attribution 3.0 Unsupported License.

significant higher SAP at both day 100- and day 130 compared to SOD1 + P (Fig. 1j). As ALS is characterized by decreased survival this is also a relevant parameter in the SOD1 G93A mouse model, but we did not find any significant difference ($p = 0.12$) between the SOD1 + P (median survival = 150 days) and SOD1 + E groups (only 1 mouse reached a neurological score of 4, and, therefore, no median survival can be calculated) (Fig. 1k). Overall, the behavioral data indicated that downregulation of CPT1 was able to delay disease progression in the SOD1 G93A mouse model based on clinical-relevant effects. This was in accordance with two other studies testing the effects of downregulating CPT1 from day 70 and until day 130 (Fig. 1l, m) and from day 100 (Supplementary Fig. 1). Sexual dimorphism have been reported to oppose a problem in the B6SJL-Tg(SOD1*G93A)1Gur/J mouse model but not in the B6.Cg-Tg(SOD1*G93A)1Gur/J model[33], which was used in this study. Accordingly, we did not find any difference in clinical-relevant phenotype between male and female SOD1 mice at day 140 nor survival (Supplementary Fig. 2).

**CPT1 antagonist restore glucose metabolism and attenuates disease processes in the spinal cord and tibialis anterior muscles in SOD1 G93A mice.** A higher level of low-density lipoprotein (LDL) and LDL/high-density lipoprotein (HDL) ratio has been associated with an increased risk of ALS[34–37]. However, discrepancies between studies assessing the effect of LDL, HDL and LDL/HDL ratio on survival in ALS exists[38,39]. To assess whether blocking CPT1 in SOD1 G93A mice affected lipid metabolism, serum levels of glucose, LDL and HDL were evaluated following termination of the animal experiment. SOD1 + P had a significant higher serum level of glucose compared to their wildtype littermates, which was not the case for SOD1 + E mice (Fig. 2a). This could indicate that etomoxir restored metabolism towards glucose utilization in the SOD1 mice. Interestingly, significant differences in the serum levels of LDL, HDL and LDL/HDL ratio were found between the SOD1 + E and SOD1 + P mice (Fig. 2b–d). SOD1 + E mice displayed a significantly lower s-LDL, higher s-HDL and lower s-LDL/HDL ratio, which is consistent with a physiological lipid phenotype for the C57Bl/6J strain and in accordance with healthy controls compared to ALS patients[37,40].

Based on the clinical-relevant effects by downregulation of CPT1 by etomoxir presented in Fig. 1, we speculated, which disease mechanisms were affected. A hallmark of disease in the SOD1 G93A model is death of motor neurons in the spinal cord. Therefore, we evaluated the serum level of neurofilament light-chain (NF-L), as this has been validated as a biomarker for neurodegenerative diseases, disease progression, survival and

death of large myelinated neurons, including motor neurons[41]. Wt mice had significantly lower serum levels of NF-L compared to both SOD1 treatment groups (Fig. 2e), indicating death of large myelinated neurons in the SOD1 G93A mouse model, as expected. However, the SOD1 + E mice had significantly lower serum levels of NF-L compared to SOD1 + P mice, which could indicate possible protective effects on large myelinated neurons following etomoxir treatment. In addition, we evaluated the protein level of choline o-acetyltransferase (ChAT) in the spinal cord to assess motor neuron death[4]. SOD1 + P mice had significantly lower levels of ChAT compared to Wt+P, which was not the case for SOD1 + E mice (Fig. 2f). This indicates that the death of motor neurons in the SOD1 + E group possibly was ameliorated by etomoxir.

Another hallmark of disease progression in ALS is reactive microglia and inflammation in the CNS[6]. Therefore, we evaluated the level of CX3CR1 in the spinal cord, as a marker of reactive microglia and activated macrophages. SOD1 + P mice had a tendency towards a significant higher level compared to Wt mice (Fig. 2g), which indicated increased pathological microglial reaction in the spinal cord of SOD1 + P mice compared to SOD1 + E mice. Next, we evaluated the levels of the anti-inflammatory cytokine interleukin (IL)-10 (Fig. 2h) and pro-inflammatory cytokines IL-1β (Fig. 2i) and tumor necrosis factor alpha (TNF-α; Fig. 2j) in the spinal cord. Both SOD1 + P and SOD1 + E mice had significantly decreased levels of IL-10 compared to Wt but etomoxir treated SOD1 mice had significantly lower levels of the pro-inflammatory cytokines compared to SOD1 + P mice (Fig. 2h–j). This indicates that etomoxir modulates the inflammatory profile in the CNS and periphery.

Subsequently, we evaluated the gene expression profile of metabolic, glial, inflammatory and oxidative stress markers in the spinal cord. Overall, SOD1 + P mice had lower expression of, *Cpt1c, Pcg1α, Glut4, Apoe, Chat, Mbp* and *Nrf2* and increased expression of *Iba1, Casp1* and *IL-17α* (Fig. 2k–m), indicating pathological processes such as mitochondrial dysfunction, down-regulated glucose metabolism, death of motor neurons, disruption of myelin, oxidative stress and inflammation. Interestingly, the CPT1 antagonist resulted in gene expressions associated with amelioration of disease such as increased mitochondrial biogenesis, glucose metabolism, myelination, oxidative stress defense and diminished inflammation (Fig. 2k–m). Knockout of *Cpt1c* has been shown to cause detrimental effects on CNS function in mice, implying that the downregulation of *Cpt1c* in the SOD1 + P group could play a role for disease progression[42,43]. *Pcg1α* was upregulated by etomoxir and has been shown to be protective in the SOD1 G93A mouse model[44]. Interestingly, etomoxir resulted

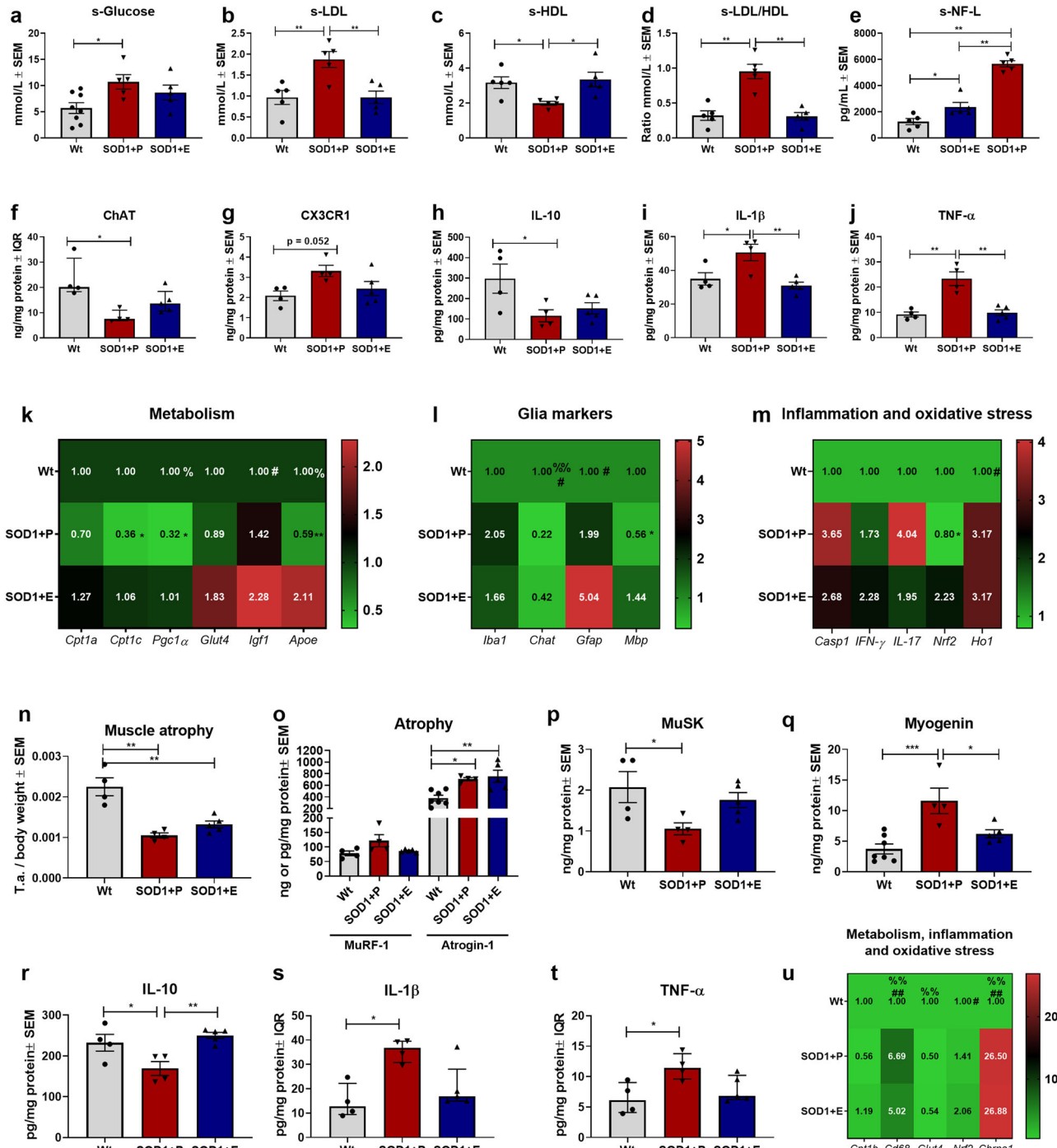

in upregulation of *Igf1* and *Apoe*, which both have been associated with protective mechanisms in ALS[45,46] and neurodegenerative diseases such as Alzheimer's disease[47] and multiple sclerosis[22]. *Glut4* was also upregulated following treatment with the CPT1 antagonist, as previously described[25].

A hallmark of disease in the SOD1 G93A model is the death of lower-motor neurons in the spinal cord as well as disruption of homeostasis in the muscles, including atrophy, denervation, inflammation, and oxidative stress[16,19,48]. Based on this we assessed the degree of muscle atrophy in the SOD1 + P and SOD1 + E compared to Wt and found that both groups had significant atrophy based on normalized weight of the tibialis anterior muscles (Fig. 2n). In addition, we investigated differences in the protein level of muscle RING-finger protein-1 (MuRF)

and atrogin-1 (Fig. 2o), which both represents markers of atrophy. Wt mice had a significant lower level of atrogin-1 in accordance with the muscle atrophy evaluation in Fig. 2n but no difference was observed between the treatment groups. Next, we evaluated the protein level of muscle skeletal receptor tyrosine-protein kinase (MuSK), which protects against denervation[49]. SOD1 + P mice had a significant decreased level compared to Wt mice this was not the case for SOD1 + E mice (Fig. 2p). Further, we evaluated the protein level of myogenin, which increases during denervation[50]. The Wt mice had significantly lower levels compared to SOD1 + P mice and the SOD1 + E mice had significantly lower levels compared to SOD1 + P mice (Fig. 2q). This could indicate that downregulation of CPT1 lipid metabolism could play a role in diminishing denervation. However, this

**Fig. 2 Downregulation of CPT1 activity by etomoxir potentially shifts metabolism towards glucose utilization and ameliorate disease mechanisms, including inflammation, mitochondrial dysfunction, and oxidative stress. a** Serum glucose levels. Mean mmol/L ± SEM. **b** Serum LDL levels. Mean mmol/L ± SEM. **c** Serum HDL levels. Mean mmol/L ± SEM. **d** Serum LDL/HDL ratio levels. Mean mmol/L ratio ± SEM. **e** Serum NF-L levels. Mean pg/mL ± SEM, **f** ChAT levels in lumbar spinal cord tissue homogenate. Median ng/mg total protein ± IQR. **g** CX3CR1 levels in lumbar spinal cord tissue homogenate. Mean ng/mg total protein ± SEM. **h** IL-10 levels in lumbar spinal cord tissue homogenate. Mean pg/mg total protein ± SEM. **i** IL-1β levels in lumbar spinal cord tissue homogenate. Mean pg/mg total protein ± SEM. **j** TNF-α levels in lumbar spinal cord tissue homogenate. Mean pg/mg total protein ± SEM. **k–m** Fold-change gene expression of metabolic, glial, inflammatory and oxidative stress genes in lumbar spinal cord tissue. Mean normalized fold-change gene expression ± SEM. **n** Weight of tibialis anterior muscle at termination. Weight of tissue were normalized to body weight and expressed as mean ± SEM. **o** MuRF1 and atrogin-1 levels in tibialis anterior tissue homogenate. Mean ng/mg total protein ± SEM. **p** MuSK levels in tibialis anterior tissue homogenate. Mean ng/mg total protein ± SEM. **q** Myogenin levels in tibialis anterior tissue homogenate. Mean ng/mg total protein ± SEM. **r** IL-10 levels in tibialis anterior tissue homogenate. Mean pg/mg total protein ± SEM. **s** IL-1β levels in tibialis anterior tissue homogenate. Mean pg/mg total protein ± SEM. **t** TNF-α levels in tibialis anterior tissue homogenate. Mean pg/mg total protein ± SEM. **u** Fold-change gene expression of metabolic, inflammatory, oxidative stress and denervation genes in tibialis anterior tissue homogenate. Mean normalized fold-change gene expression ± SEM. Serum samples and tissue were obtained at termination. All data was analyzed using one-way ANOVA followed by Tukey post hoc test or Kruskal–Wallis test followed by Dunns post hoc test. $N = 5$–8 for serum analysis and 3–5 for all other experiments. Data are representative of one experiment. Gene expression was normalized to *β-actin* and *Gapdh*. *Significant differences between groups in all analyses except gene expression experiments. *$p \leq 0.05$; **$p \leq 0.01$. Significant annotations in gene expression experiments (one sign = $p \leq 0.05$, two signs = $p \leq 0.01$). * = SOD1 + P vs. SOD1 + E, % = Wt+P vs. SOD1 + P, # = Wt+P vs. SOD1 + E. Wt = wildtype, SOD1 = SOD1 G93A genotype, E = etomoxir, P = placebo, SEM = standard error of mean, IQR = interquartile range, LDL = low-density lipoproteins, HDL = High-density lipoproteins, NF-L = Neurofilament light-chain, ChAT = Choline o-acetyltransferase, MuSK = Muscle skeletal receptor tyrosine-protein kinase, MuRF1 = Muscle RING-finger protein-1.

will require further investigations. One of the mechanisms causing increased muscle pathology is inflammation. Therefore, we also evaluated the levels of IL-10, IL-1β and TNF-α in the tibialis anterior muscles. SOD1 + P mice had decreased levels of IL-10 (Fig. 2r) and increased levels of IL-1β (Fig. 2s) and TNF-α (Fig. 2t) compared to Wt and SOD1 + E mice. Finally, we evaluated the gene expression profile of metabolic, inflammatory, oxidative stress and denervation markers in the muscles (Fig. 2u). SOD1 + P and SOD1 + E mice had significantly increased levels of the active macrophage marker *Cd68*. In addition, both SOD1 groups had significantly downregulated *Glut4* expression. SOD1 + E mice had significantly increased expression of the oxidative stress defense gene *Nrf2*, indicating that etomoxir potentially diminished oxidative stress. Finally, we assessed the expression of *Chrna1*, a marker of denervation, and found that both SOD1 groups had increased expression indicating denervation. In summary, this indicate that etomoxir was able to alleviate the disease progression and diminish disease processes such as inflammation and oxidative stress to some extent but not in the terminal stage. The SOD1 G93A mouse model is primarily characterized by lower-motor neuron disease pathology, however, this and other studies indicate that *Mbp* and *Cd68* might also be affected in the brain[51] (Supplementary Fig. 3).

**Downregulation of CPT1 lipid metabolism modulates CPT1A and CPT1C labeling, myelination, astrogliosis, reactive microglia, and motor neuron survival in the spinal cord in SOD1 G93A mice.** Following the clinical-relevant effects of downregulating CPT1 lipid metabolism by etomoxir in the SOD1 G93A mouse model (Fig. 1) and the changes in metabolic, inflammatory and oxidative stress markers in the lumbar spinal cord (Fig. 2), we performed an immunohistochemical peroxidase staining on the lumbar spinal cord to evaluate morphological changes. To evaluate the effects of CPT1 downregulation, SOD1 G93A mice received etomoxir (SOD1 + E) or placebo (SOD1 + P) from day 70 and until termination at day 130. SOD1 + E mice had significantly lower neuroscore and increased grip strength compared to SOD1 + P mice (Fig. 1l, m). First, we evaluated whether etomoxir affected the labeling of CPT1A. We found that SOD1 + P mice had increased labeling compared to SOD1 + E and Wt (Fig. 3a–c and Supplementary Fig. 4), which indicated that etomoxir decreased the level of CPT1A, as previously described[23]. Next, we evaluated whether etomoxir affected the

labeling of CPT1C. However, we did not find any indications of difference in the labeling (Fig. 3d, e and Supplementary Fig. 4). Previously, we and others have shown that CPT1 downregulation results in diminished demyelination[13,20]. Further, demyelination has been observed in the SOD1 G93A mouse model- and ALS patients[51,52]. Therefore, we performed staining for MBP. In agreement with the gene expression studies, SOD1 + P mice were found to have decreased labeling of MBP, which was counteracted by etomoxir (Fig. 3g–i and Supplementary Fig. 4).

Spinal cords from ALS patients and SOD1 mouse model are characterized by astrogliosis and increased reactive astrocytes[53]. Therefore, we assessed the labeling of the reactive astrocyte marker, GFAP, in the spinal cord of SOD1 mice. SOD1 + P mice had increased labeling of GFAP compared to Wt and SOD1 + E mice (Fig. 4a–c and Supplementary Fig. 4), indicating increased reactive astrocytes in the SOD1 + P group. Another hallmark of disease in ALS and the SOD1 G93A model is neuroinflammation[1]. Therefore, we evaluated whether etomoxir altered the labeling of IBA1, a marker of reactive microglia and macrophages, in the lumbar spinal cord. SOD1 + E mice had decreased labeling compared with SOD1 + P mice (Fig. 4d–f), which was in accordance with the inflammatory data presented in Fig. 2. Finally, we assessed the labeling of ChAT, a marker of cholinergic neurons, in the ventral horn in the lumbar spinal cord and found indications of decreased labeling in SOD1 + P compared to SOD1 + E mice (Fig. 4g–i). Overall, the immunohistochemical staining corresponded to the data presented in Figs. 1 and 2.

**Genetic downregulation of CPT1A activity delays disease onset and alleviate clinical symptoms in SOD1 G93A mice.** Based on the effect of the CPT1 antagonist in the SOD1 G93A model and our previous results showing that *Cpt1a$^{P479L/P479L}$* mutated mice are resistant to the experimental autoimmune encephalomyelitis (EAE) model[20], we tested the effect of crossing *Cpt1a$^{P479L/P479L}$* mutated female mice with SOD1 G93A male mice. The *Cpt1a$^{P479L/P479L}$* mouse strain was generated by mutating the genomic *Cpt1a* gene, causing no expression changes in different tissues[20]. The mutation results in a CPT1A activity of ~22% compared to the wildtype protein in its' homozygote form and ~70% activity in the heterozygote form[54]. SOD1 G93A mice with a heterozygote *Cpt1a$^{Wt/P479L}$* (SOD1$^{Wt/Cpt1a}$) and SOD1 G93A mice with a homozygote *Cpt1a$^{P479L/P479L}$* mutation (SOD1$^{Cpt1a/Cpt1a}$)

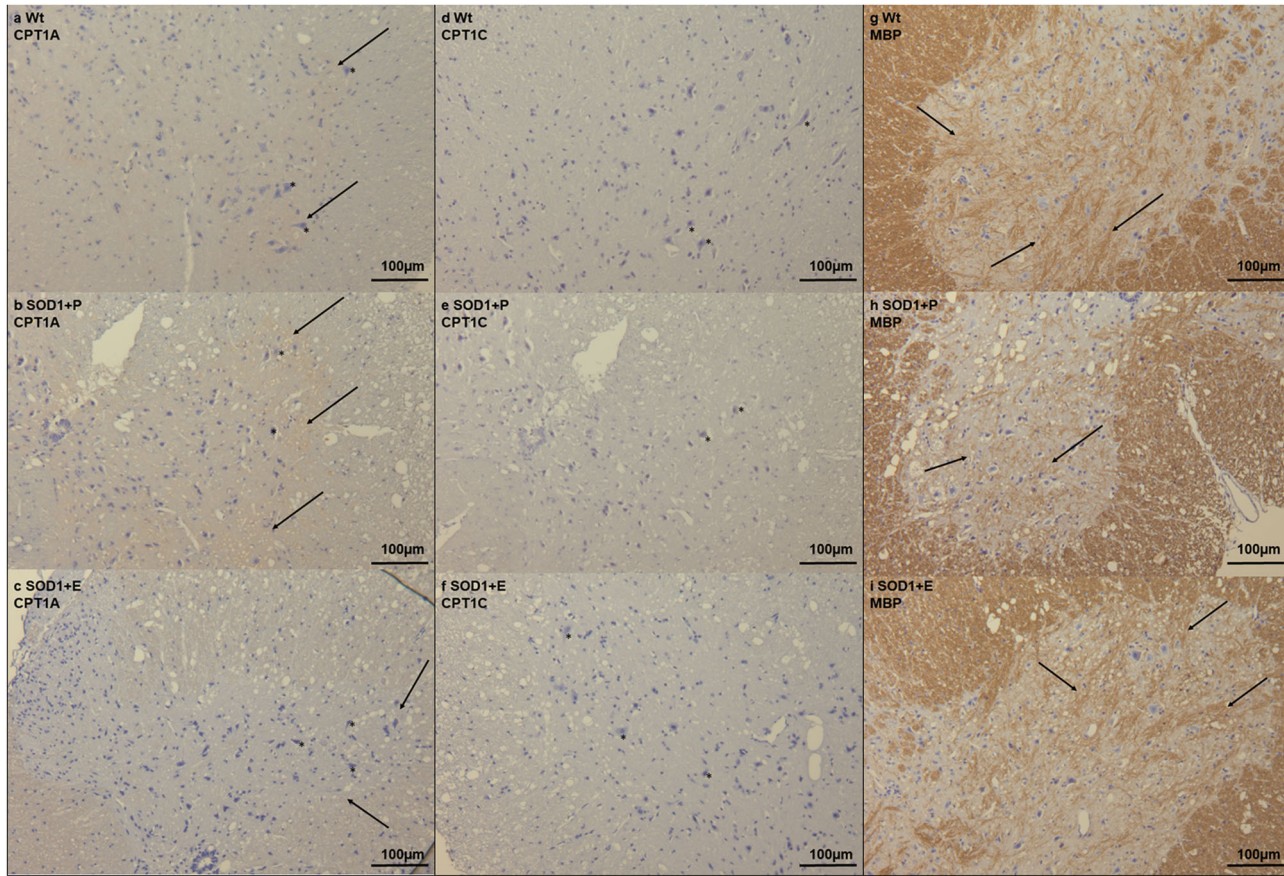

**Fig. 3 Immunohistochemical staining in the lumbar spinal from the SOD1 G93A etomoxir experiment. a–c** CPT1A staining in lumbar spinal cord from Wt, SOD1 + P and SOD1 + E mice at day 130 indicating increased labeling in SOD1 + P mice (arrows) and pathological morphology of neurons (asterisks). **d–f** CPT1C staining in lumbar spinal cord from Wt, SOD1 + P and SOD1 + E mice at day 130 indicating no difference in the labeling in SOD1 + P mice but differences in the morphology of neurons (asterisks). **g–i** MBP staining in lumbar spinal cord from Wt, SOD1 + P and SOD1 + E mice at day 130 indicating decreased labeling in SOD1 + P mice (arrows). All images are presented with 16x magnification. N = 2–4 animals per group. WT = wild-type, SOD1 = SOD1 G93A genotype, E = etomoxir, P = placebo.

were bred. The female SOD1$^{Wt/Cpt1a}$ and SOD1$^{Cpt1a/Cpt1a}$ clinical disease phenotypes were compared to female SOD1 G93A mice without Cpt1a mutation (SOD1) (Fig. 5a).

The groups were tested weekly from day 70 by neurological score, grip strength- and hangwire tests as previously described[24]. The SOD1$^{Wt/Cpt1a}$ (median onset = 97 days) and SOD1$^{Cpt1a/Cpt1a}$ (median onset = 103 days) had significantly (p = 0.0014) later onset of tremor compared to SOD1 (median onset = 93 days) (Fig. 5b). The SOD1$^{Wt/Cpt1a}$ and SOD1$^{Cpt1a/Cpt1a}$ had significantly lower neurological score, higher latency to fall on the hangwire test and normalized grip strength at several time points compared to SOD1 (Fig. 5c–e). No significant differences were observed on weight (Fig. 5f). Furthermore, we evaluated the spontaneous activity in the cylinder test and found that both SOD1$^{Wt/Cpt1a}$ and SOD1$^{Cpt1a/Cpt1a}$ had significantly higher activity at day 70- and day 100 compared to SOD1 mice (Fig. 2g). Based on the effects of downregulating CPT1 by the irreversible CPT1 antagonist on visuospatial memory, we speculated whether the decreased CPT1A activity affected memory in the mice with Cpt1a mutations. Interestingly, SOD1$^{Cpt1a/Cpt1a}$ mice had a significant higher visuospatial memory at day 70-, 100-, and day 130 compared to SOD1 mice (Fig. 5i). The SOD1$^{Wt/Cpt1a}$ mice had significant higher visuospatial memory compared to SOD1 mice at day 100 and day 130 (Fig. 5i). Next, we evaluated whether the downregulated CPT1A activity affected survival. Remarkably, SOD1$^{Wt/Cpt1a}$ (median survival = 152) and

SOD1$^{Cpt1a/Cpt1a}$ (median survival = 153) had significantly (p = 0.032) increased survival compared to SOD1 mice (median survival = 147) (Fig. 5j). We repeated the SOD1 x Cpt1a P479L mutant experiment and evaluated neurological score until day 132 and found similar results (Fig. 5k). Finally, we evaluated the survival in male SOD1$^{Cpt1a/Cpt1a}$ (median survival = 160 days) and SOD1 mice (median survival = 149 days) and found significant differences (p = 0.0016) according to the results from the females (Fig. 5l). In general, the behavioral data was consistent with the CPT1 antagonist study (Fig. 1). However, minor effects were observed on the hangwire- and grip strength tests. This is consistent with the fact that Cpt1a P479L hetero- or homozygote mutations do not affect the activity of CPT1B. In general, the Cpt1a P479L mutation delayed disease progression and thereby confirmed the results from the CPT1 antagonist study.

**Cpt1a P479L mutation restore glucose metabolism and attenuates disease processes in the spinal cord and tibialis anterior muscles in SOD1 G93A mice.** In conjunction with the CPT1 antagonist experiment, we assessed the effect of the Cpt1a$^{Wt/P479L}$ and Cpt1a$^{P479L/P479L}$ mutation on the serum levels of glucose (Fig. 6a), LDL (Fig. 6b), HDL (Fig. 6c), and LDL/HDL ratio (Fig. 6d) in SOD1 mice. Interestingly, we found that downregulation of CPT1A activity in SOD1 mice increased

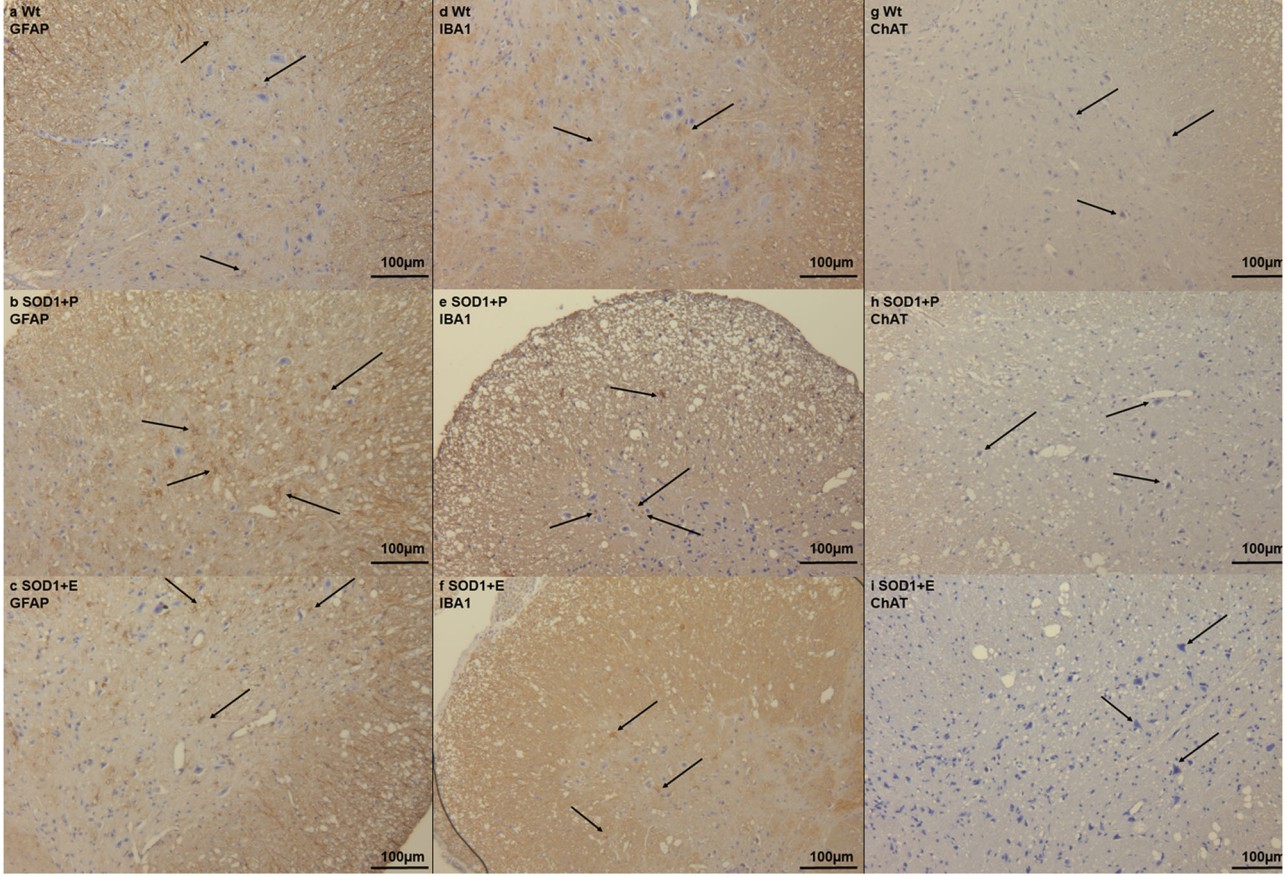

**Fig. 4 Immunohistochemical staining in the lumbar spinal from the SOD1 G93A etomoxir experiment. a–c** GFAP staining in lumbar spinal cord from Wt, SOD1 + P and SOD1 + E mice at day 130 indicating increased number of reactive astrocytes in the ventral horn in SOD1 + P mice (arrows) compared to Wt and SOD1 + E mice. **d–f** IBA1 staining in lumbar spinal cord from Wt, SOD1 + P and SOD1 + E mice at day 130, indicating increased labeling and infiltration of reactive microglia in the ventral horn in SOD1 + P mice (arrows) compared to SOD1 + E and Wt mice. **g–i** ChAT staining in lumbar spinal cord from Wt, SOD1 + P and SOD1 + E mice at day 130 indicating decreased labeling in SOD1 + P mice (arrows) and pathological morphology of neurons. All images are presented with x16 magnification. N = 2–4 animals per group. WT = wild-type, SOD1 = SOD1 G93A genotype, E = etomoxir, P = placebo.

glucose metabolism and restored lipoprotein levels towards a reduced risk of ALS.

Based on the clinical-relevant effects by downregulation of CPT1A activity by genetic mutation (Fig. 5) and the effects of etomoxir (Figs. 1, 2), we speculated whether the serum level of NF-L was affected by the mutation. SOD1$^{Cpt1a/Cpt1a}$ mice had decreased levels of NF-L in serum compared to SOD1 mice but no significant difference was detected (Fig. 6e). In addition, we evaluated the protein level of ChAT in the spinal cord to assess motor neuron death[4]. SOD1$^{Cpt1a/Cpt1a}$ mice had significantly higher levels of ChAT compared to SOD1 mice (Fig. 6f). This indicates that the death of motor neurons was attenuated by the downregulated CPT1A activity due to the P479L mutation.

In addition, we evaluated whether the downregulated CPT1A activity affected inflammation as etomoxir did (Fig. 2). Therefore, we evaluated the level of CX3CR1 in the spinal cord, as a marker of reactive microglia and activated macrophages. Both SOD1$^{Wt/Cpt1a}$ and SOD1$^{Cpt1a/Cpt1a}$ had decreased levels compared to SOD1 mice (Fig. 6g), but it did not reach significance. Next, we evaluated the levels of the anti-inflammatory cytokine IL-10 (Fig. 6h) and pro-inflammatory cytokines IL-1β (Fig. 6i) and TNF-α (Fig. 6j) in the spinal cord. SOD1$^{Cpt1a/Cpt1a}$ had significantly increased levels of IL-10 and both SOD1$^{Cpt1a/Cpt1a}$ and SOD1$^{Wt/Cpt1a}$ had decreased levels of TNF-α compared to SOD1 but no difference was found in IL-1β levels. This indicates

that the downregulated CPT1A activity modulates the inflammatory profile in the CNS.

Subsequently, we evaluated the gene expression profile of metabolic, glial, inflammatory and oxidative stress markers in the spinal cord. Overall, SOD1 mice had lower expression of, *Cpt1c*, *Pcg1α*, *Glut4*, *Apoe*, *Chat*, *Mbp*, *Nrf2* and *Ho1* increased expression of *Casp1* (Fig. 6k–m), indicating pathological processes such as mitochondrial dysfunction, downregulated glucose metabolism, death of motor neurons, disruption of myelin, oxidative stress and inflammation as in the etomoxir study. Interestingly, the downregulated CPT1A activity resulted in gene expressions associated with amelioration of disease such as increased mitochondrial biogenesis, glucose metabolism, survival of ChAT expressing neurons, myelination, oxidative stress defense and diminished inflammation in the SOD1$^{Wt/Cpt1a}$ and SOD1$^{Cpt1a/Cpt1a}$ mice (Fig. 6k–m). This indicates that the genetic downregulation of CPT1A activity had similar effects as the pharmacological downregulation of CPT1A and CPT1B activity by etomoxir.

As previously described another hallmark of disease in the SOD1 G93A model is the disruption of homeostasis in the muscles including atrophy, denervation, inflammation, and oxidative stress[16,19,48]. Based on the effects seen by etomoxir treatment (Fig. 2n–t), we evaluated the effects of downregulating CPT1A activity on muscle pathogenic processes in the tibialis anterior muscles. SOD1$^{Wt/Cpt1a}$ and SOD1$^{Cpt1a/Cpt1a}$ mice had a

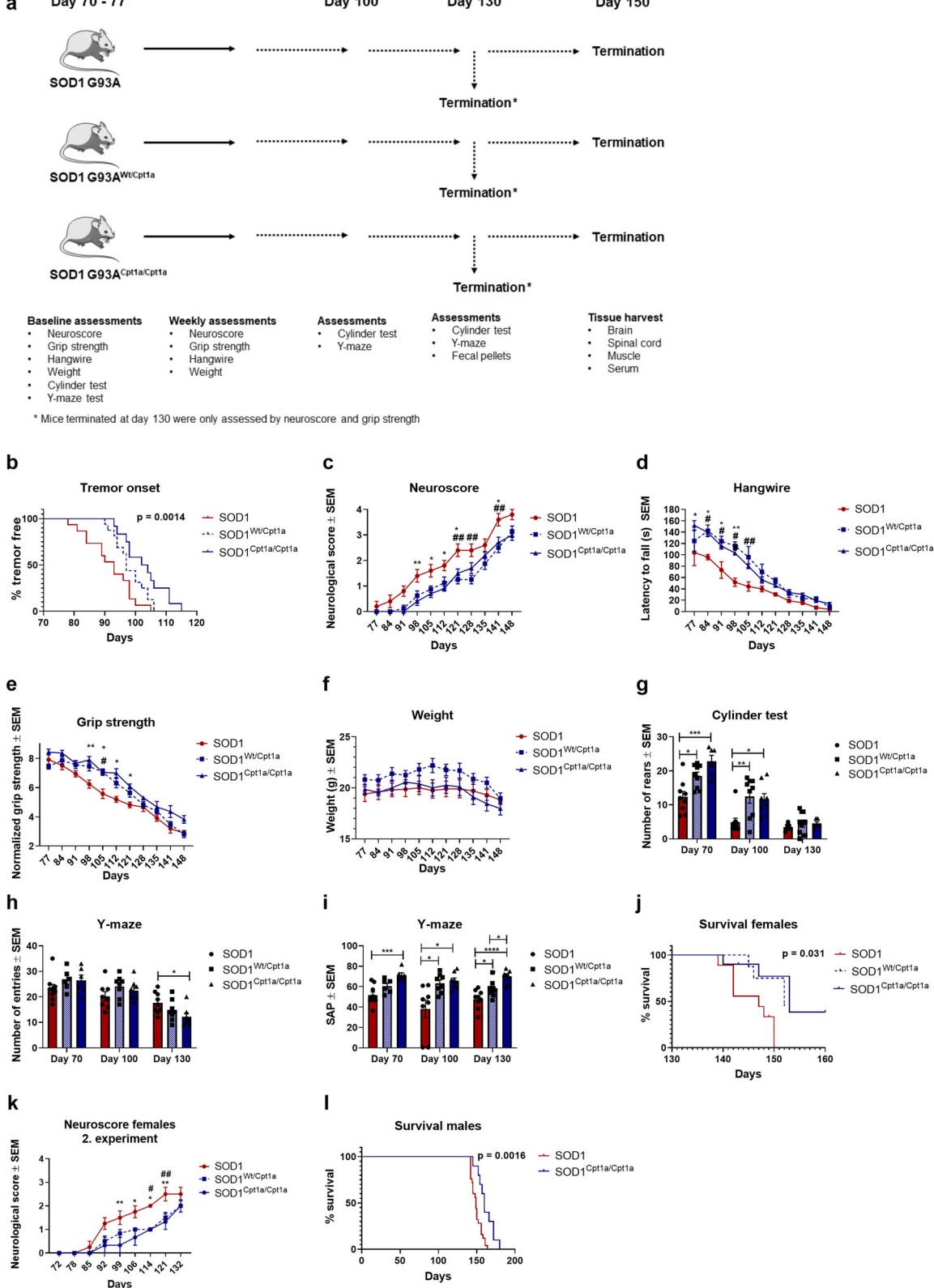

lower degree of muscle atrophy compared to SOD1 mice (Fig. 6n), but the difference was not significant. In addition, SOD1$^{Wt/Cpt1a}$ and SOD1$^{Cpt1a/Cpt1a}$ mice had lower levels of MuRF1 (Fig. 6o) in accordance with the higher normalized weight of muscle tissue (Fig. 6n). Further, SOD1$^{Wt/Cpt1a}$ mice had a tendency towards a significantly lower level of the atrophy marker atrogin-1[16] compared to SOD1 mice (Fig. 6o). Next, we evaluated the

protein level of MuSK and found that SOD1$^{Cpt1a/Cpt1a}$ mice had higher levels compared to SOD1 mice (Fig. 6p). In addition, SOD1$^{Wt/Cpt1a}$ and SOD1$^{Cpt1a/Cpt1a}$ mice had significantly lower protein levels of myogenin, which could indicate a lower degree of denervation in these mice (Fig. 6q). However, this will require further experiments to confirm. As in the spinal cord, inflammation plays a central role in the progression of disease in the

**Fig. 5 Genetic inhibition of CPT1A ameliorates clinical symptoms and delays disease progression in the SOD1 G93A mouse model. a** Experimental setup, mice were evaluated with behavioral tests from day 70 (baseline) and tested weekly until day 150. **b** Time point for onset of disease defined as visible tremor in hindlimbs, log-rank survival analysis. **c** Neuroscore was assessed once a week. Mean neuroscore ± SEM. **d** Hangwire test was performed once a week. Mean latency to fall of the grid ± SEM. **e** Grip strength was measured once a week from day 70. Data are presented as mean normalized strength ± SEM. **f** Mean body weight expressed in grams ± SEM. **g** Cylinder test was performed at day 70, 100, and 130. Mean rears ± SEM. One-way ANOVA with Tukey post hoc test. **h** Y-maze test was conducted at day 70, 100, and 130. Mean number of entries ± SEM. One-way ANOVA with Tukey post hoc test **i** Y-maze test was conducted at day 70, 100, and 130. Mean spontaneous alternation percentage ± SEM. One-way ANOVA with Tukey post hoc test. **j** Survival analysis for females, survival was defined as neuroscore below 4, log-rank test. **k** Mean neuroscore for SOD1 mice evaluated from day 70 and until day 132 ± SEM. **l** Survival analysis for females, survival was defined as neuroscore below 4, log-rank test. Animals were tested weekly from day 70 of age until day 148. $N = 5$–10, except for tremor onset ($n = 12$–16) and survival analyses ($n = 8$–10 for females, $n = 10$–25). Data are representative of two independent animal experiments for females from day 70 and until day 150 and one for day 70 and until day 132. If nothing else is noted, data was analyzed using repeated measure two-way ANOVA followed by Tukey post hoc test. *Significant differences between SOD1$^{Cpt1a/Cpt1a}$ and SOD1, #significant differences between SOD1$^{Wt/Cpt1a}$ and SOD1 in behavioral tests. *$p \leq 0.05$; **$p \leq 0.01$; ***$p \leq 0.001$, ****$p \leq 0.0001$. SOD1 = SOD1 G93A genotype, SOD1$^{Wt/Cpt1a}$ = SOD1 G93A mice with heterozygote *Cpt1a P479L* mutation, SOD1$^{Cpt1a/Cpt1a}$ = SOD1 G93A mice with homozygote *Cpt1a P479L* mutation. Mouse figure used in experimental setup figures were obtained from; https://smart.servier.com/ and used according to the Creative Commons Attribution 3.0 Unsupported License.

muscles. Therefore, we also evaluated the levels of IL-10, IL-1β and TNF-α in the tibialis anterior muscles. SOD1$^{Cpt1a/Cpt1a}$ mice had increased levels of IL-10 (Fig. 6r) and both SOD1$^{Wt/Cpt1a}$ and SOD1$^{Cpt1a/Cpt1a}$ mice had decreased levels of TNF-α (Fig. 6t). No difference was found in the levels of IL-1β between the groups (Fig. 6s). This indicates that the downregulation of CPT1A activity attenuated inflammation in the tibialis anterior muscles.

Finally, we evaluated the gene expression profile of inflammatory, oxidative stress and denervation markers in the muscles (Fig. 6u). SOD1$^{Wt/Cpt1a}$ and SOD1$^{Cpt1a/Cpt1a}$ mice had decreased expression of *Cd68* compared to SOD1 mice (Fig. 6u) in accordance with the decreased level of inflammatory cytokines in the muscles (Fig. 6r–t). SOD1$^{Wt/Cpt1a}$ and SOD1$^{Cpt1a/Cpt1a}$ mice had decreased expression of the oxidative defense gene *Nrf2* compared to SOD1 mice (Fig. 6u), which could indicate that oxidative stress levels were lower or that *Nrf2* upregulation was impaired. Finally, we assessed the expression of *Chrna1* and found that both SOD1$^{Wt/Cpt1a}$ and SOD1$^{Cpt1a/Cpt1a}$ had decreased expression compared to SOD1 mice (Fig. 6u), which indicates that the downregulated CPT1A activity possibly ameliorated denervation. In summary, these findings indicate that downregulation of CPT1A activity was able to alleviate the disease progression and diminish disease processes such as inflammation, oxidative stress, decrease the death of ChAT expressing neurons and ameliorate denervation to some extent.

**Upregulation of CPT1 activity by 60% saturated high-fat diet accelerates disease progression in the SOD1 G93A mouse model.** Downregulation of CPT1 isoforms A and B activity by etomoxir (Figs. 1, 2) and CPT1A activity by the *Cpt1a* P479L mutation (Figs. 5, 6) delayed disease progression and attenuated multiple pathological mechanisms including inflammation and oxidative stress in the SOD1 G93A mouse model. Based on this we hypothesized that the upregulation of CPT1A and CPT1B activity would exacerbate disease progression and severity. High-fat diets (HFD) with a high level of saturated fatty acids increases the level of acyl-CoA[55], CPT1A[56] and CPT1B activity[57] and thereby β-oxidation in the periphery, including liver and muscle tissue. In addition, HFD increases pyruvate dehydrogenase kinase-4 (PDK4) activity and thereby downregulates the pyruvate dehydrogenase complex and thus the conversion of pyruvate into acetyl-CoA to be used in the Krebs cycle[58]. HFD also down-regulates the glucose transporters within the CNS and increases insulin resistance in the periphery and CNS, which potential results in decreased glucose metabolism[59]. Previous studies on the effects of HFD in in vivo models of ALS and ALS patients have resulted in inconsistent conclusions[60–65] due to large differences

in the composition of the diets[61]. Therefore, we tested whether a western HFD, 60% saturated fats, affected the disease progression in the SOD1 G93A model.

SOD1 G93A mice were fed with a HFD (SOD1 + HFD) (Fig. 7a) and compared with SOD1 G93A mice fed a normal diet (ND) (SOD1 + ND). SOD1 + HFD had a significant ($p = 0.0052$) earlier onset of tremor (median onset = 91 days) compared to SOD1 + ND (median onset = 98 days), loss of normalized grip strength and lower latency to fall on the hangwire test (Fig. 7b–e). Interestingly, wildtype mice receiving HFD (Wt + HFD) also had a significant decrease in normalized grip strength compared to wildtype mice receiving ND (Wt + ND) (Fig. 7d). HFD resulted in a significant weight gain for both Wt + HFD and SOD1 + HFD compared to their respective controls (Fig. 7f). HFD resulted in a significantly lower activity in the cylinder test (Fig. 7g). SOD1 mice had a significantly lower number of entries in the y-maze, which is in accordance with motor neuron degeneration (Fig. 7h). Both Wt+HFD and SOD1 + HFD had a significant decrease in their visuospatial memory based on the y-maze test in agreement with previous studies in Wt mice receiving HFD (Fig. 7i). Finally, we evaluated whether HFD affected survival. We found that SOD1 + HFD (median survival = 140) had significantly ($p = 0.0083$) decreased survival compared to SOD1 + ND (median survival = 147). Overall, our data indicated that HFD results in a more severe disease phenotype in the SOD1 G93A model. Differences seen from other studies using HFD could possibly be attributed to genetic background and differences in the composition of the HFD[9,66,67].

**60% HFD results in hyperglycemia, decreased glucose metabolism, and exacerbates disease processes in the spinal cord and tibialis anterior muscles in SOD1 G93A mice.** Based on the exacerbation of clinical-relevant phenotype in SOD1 mice fed a 60 % HFD, we assessed the serum levels of glucose, LDL, HDL, and LDL/HDL ratio. SOD1 + HFD had significantly increased glucose levels compared to SOD1 + ND and Wt + ND (Fig. 8a), indicating hyperglycemia, possibly due to decreased glucose metabolism. In addition, SOD1 + HFD had significant differences in the level of LDL (Fig. 8b), HDL (Fig. 8c), and LDL/HDL ratio (Fig. 8d), which were in accordance with the differences seen for the SOD1 + P group (Fig. 2b–d). This indicated that 60 % HFD resulted in downregulated glucose utilization and increased lipoprotein levels in the serum, pointing towards metabolic alternations.

Next, we evaluated the protein level of ChAT in the spinal cord to assess whether HFD affected motor neuron death[4]. ChAT was significantly decreased in both SOD1 + ND and SOD1 + HFD

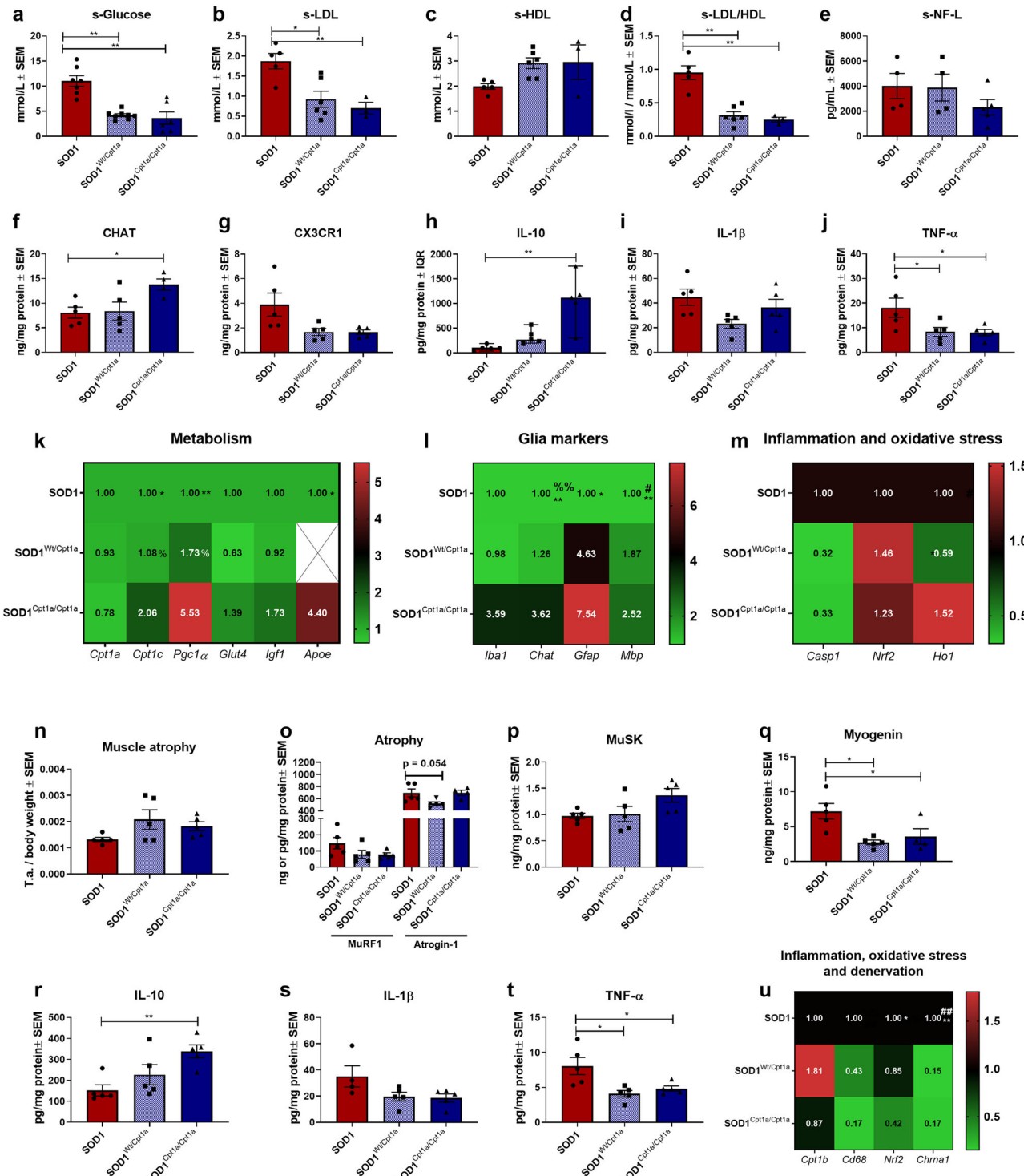

compared to Wt+ND (Fig. 8e). However, the SOD1 + HFD mice had the lowest level of ChAT in the spinal cord, possibly indicating higher disease activity.

In addition, we evaluated whether the upregulated CPT1 activity due to 60 % HFD affected inflammation in the spinal cord. In general, SOD1 mice had decreased levels of IL-10 (Fig. 8f) and increased levels of IL-1β (Fig. 8g) and TNF-α (Fig. 8h). SOD1 + HFD mice had significantly increased levels of IL-1β compared to SOD1 + ND mice (Fig. 8g) indicating possibly increased inflammation. Following these findings, we assessed changes in the gene expression profile of metabolic, glial,

inflammatory and oxidative stress markers in the spinal cord. study. Interestingly, HFD resulted in changes in expression of Cpt1a, Cpt1c, Pcg1α, Iba1, Chat, Gfap, Cd68, Ho1, Nox2 and Nrf2 (Fig. 8i, j). This indicates that in general, HFD results in changes associated with increased disease activity such as decreased mitochondrial biogenesis, death of motor neurons, inflammation and decreased defense against oxidative stress in the spinal cord.

We previously found that downregulation of CPT1 activity attenuated pathological mechanisms in the tibialis anterior muscle (Figs. 2 and 6). Therefore, we evaluated whether the upregulation of CPT1B due to 60% HFD had any detrimental

**Fig. 6 Downregulation of CPT1A activity by *Cpt1a* P479L mutations potentially shifts metabolism towards glucose utilization and ameliorate disease mechanisms including inflammation, mitochondrial dysfunction, and oxidative stress. a** Serum glucose levels. Mean mmol/L ± SEM. **b** Serum LDL levels. Mean mmol/L ± SEM. **c** Serum HDL levels. Mean mmol/L ± SEM. **d** Serum LDL/HDL ratio levels. Mean mmol/L ratio ± SEM. **e** Serum NF-L levels. Mean pg/mL ± SEM. **f** ChAT levels in lumbar spinal cord tissue homogenate. Mean ng/mg total protein ± SEM. **g** CX3CR1 levels in lumbar spinal cord tissue homogenate. Mean ng/mg total protein ± SEM. **h** IL-10 levels in lumbar spinal cord tissue homogenate. Median pg/mg total protein ± IQR. **i** IL-1β levels in lumbar spinal cord tissue homogenate. Mean pg/mg total protein ± SEM. **j** TNF-α levels in lumbar spinal cord tissue homogenate. Mean pg/mg total protein ± SEM. **k–m** Fold-change gene expression of metabolic, glial, inflammatory and oxidative stress genes in lumbar spinal cord tissue. Mean normalized fold-change gene expression ± SEM. **n** Weight of tibialis anterior muscle at termination. Weight of tissue were normalized to body weight and expressed as mean ± SEM. **o** MuRF1 and atrogin-1 levels in tibialis anterior tissue homogenate. Mean ng/mg total protein ± SEM. **p** MuSK levels in tibialis anterior tissue homogenate. Mean ng/mg total protein ± SEM. **q** Myogenin levels in tibialis anterior tissue homogenate. Mean ng/mg total protein ± SEM. **r** IL-10 levels in tibialis anterior tissue homogenate. Mean pg/mg total protein ± SEM. **s** IL-1β levels in tibialis anterior tissue homogenate. Mean pg/mg total protein ± SEM. **t** TNF-α levels in tibialis anterior tissue homogenate. Mean pg/mg total protein ± SEM. **u** Fold-change gene expression of metabolic, inflammatory, oxidative stress and denervation genes in tibialis anterior tissue homogenate. Mean normalized fold-change gene expression ± SEM. Serum samples and tissue were obtained at termination. All data was analyzed using one-way ANOVA followed by Tukey post hoc test or Kruskal–Wallis test followed by Dunns post hoc test. $N = 3–8$ for serum analysis and 3–5 for all other experiments. Data are representative of one experiment. Gene expression was normalized to β-actin and *Gapdh*. *Significant differences between groups in all analyses except gene expression experiments. *$p ≤ 0.05$; **$p ≤ 0.01$. Significant annotations in gene expression experiments (one sign = $p ≤ 0.05$, two signs = $p ≤ 0.01$). * = SOD1 vs. SOD1$^{Cpt1a/Cpt1a}$, % = SOD1$^{Wt/Cpt1a}$ vs. SOD1$^{Cpt1a/Cpt1a}$, # = SOD1 vs. SOD1$^{Wt/Cpt1a}$. SOD1 = SOD1 G93A genotype, SOD1$^{Wt/Cpt1a}$ = SOD1 G93A mice with heterozygote *Cpt1a* P479L mutation, SOD1$^{Cpt1a/Cpt1a}$ = SOD1 G93A mice with homozygote *Cpt1a* P479L mutation, SEM = standard error of mean, IQR = interquartile range, LDL = low-density lipoproteins, HDL = High-density lipoproteins, NF-L = Neurofilament light-chain, ChAT = Choline o-acetyltransferase, MuSK = Muscle skeletal receptor tyrosine-protein kinase, MuRF1 = Muscle RING-finger protein-1.

effects in the tibialis anterior muscle. 60% HFD resulted in increased muscle atrophy (Fig. 8l) and decreased levels of MuSK (Fig. 8m), indicating increased denervation. Unexpectedly, we did not observe an increase in the level of MuRF1 in SOD1 + HFD mice (Fig. 8n). Additionally, we evaluated whether the diet affected the level of anti- and pro-inflammatory cytokines in the tibialis anterior muscle. SOD1 + HFD mice had a tendency to lower level of IL-10, which were however not statistical significant (Fig. 8o) no changes were observed in IL-1β (Fig. 8p) and TNF-α level (Fig. 8q). Finally, we investigated whether 60% HFD resulted in changes in gene expression in the muscles. HFD resulted in increased β-oxidation in the muscle, inflammation, and decreased glucose metabolism (Fig. 8r), which is consistent with previous studies. In general, the gene expression profiles in the muscle tissue were accordant with those in the spinal cord. Interestingly, *Cpt1c* gene expression was found to be decreased in the brain following HFD in both Wt and SOD1 G93A mice (Supplementary Fig. 3). Knockout of *Cpt1c* decreases cognitive function[42,43], which could indicate that the decreased *Cpt1c* expression in the HFD groups play a role in the decreased performance in the Y-maze test (Fig. 7i).

In summary, these findings indicate that upregulation of CPT1A and CPT1B activity resulted in a more aggressive clinical-relevant behavior and aggravated disease processes such as inflammation, oxidative stress, increased the death of ChAT expressing neurons and possibly muscle atrophy and denervation.

**Upregulation of CPT1 activity by corticosterone accelerates disease progression in the SOD1 G93A mouse model.** Any imbalances to an organism's homeostasis elicit a complex stress response that causes activation of the neuroendocrine and autonomic system. One of the essential systems in the stress response is the hypothalamic-pituitary-adrenal (HPA) axis[68]. During acute stress, such as critical sickness, the stress response is beneficial for survival. However, prolonged stress, due to psychological or physiological circumstances, causes an overactivation of the HPA-axis resulting in chronic high levels of glucocorticoids (cortisol). High levels of cortisol, in turn, result in increased activity of PDK4, which downregulates glucose metabolism[69]. In addition, cortisol promotes insulin resistance, which forces metabolism towards lipolysis[70,71] and increases β-oxidation[72]. These findings could indicate that increased levels of

glucocorticoids could increase the activity of CPT1. Several studies support the importance of a dysregulation of the HPA-axis in ALS and the SOD1 G93A mouse model. ALS patients have been found to have increased morning cortisol blood levels and a disruption of the cortisol awakening response[73,74]. Studies have shown that prolonged high levels of cortisol leads to glucocorticoid receptor resistance, which eventually results in failure to downregulate the inflammatory response[75]. Chronic restraint stress in SOD1 G93A mice increased the level of corticosterone, which correlated with decreased survival and increased inflammation[76].

Based on this, we hypothesized that increased levels of glucocorticoids would result in a downregulation of glucose metabolism and an upregulation of CPT1-mediated lipid metabolism. Therefore, we tested the effect of administering the mouse equivalent of cortisol, corticosterone, to SOD1 G93A mice from day 70 and until day 100 (Fig. 9a). SOD1 G93A mice receiving corticosterone (SOD1 + CORT) had significant ($p = 0.03$) earlier onset of tremor (median onset = 79 days) compared to SOD1 + V (median onset = 87 days), higher neurological score, decreased latency to fall and decreased normalized grip strength (Fig. 9b–e). Interestingly, also wildtype mice receiving corticosterone (Wt + CORT) had a significant decrease in normalized grip strength (Fig. 9e). SOD1 G93A + CORT and Wt + CORT had a significant increase in weight (Fig. 9f), which could indicate changes in metabolism, according to the previously presented literature. Further, SOD1 + CORT had a significant reduction in activity in the cylinder test (Fig. 9g) and visuospatial memory in the y-maze test (Fig. 9h).

We terminated a group of mice at day 100 to evaluate whether the oral administration of CORT affected serum CORT levels. Serum levels of CORT was significantly increased in the SOD1 + CORT mice at day 100 compared to SOD1 + V (Fig. 9i), in accordance with previous studies using chronic restraint stress in SOD1 mice[76]. Next, we examined whether CORT affected glucose levels in serum as an indication of down-regulated glucose metabolism. SOD1 + CORT mice had higher serum glucose levels (Fig. 9j), indicating downregulated utilization of glucose.

Finally, we evaluated changes in gene expression of metabolic, inflammatory and oxidative stress genes in spinal cord and tibialis anterior tissue obtained at day 130. SOD1 + CORT mice had

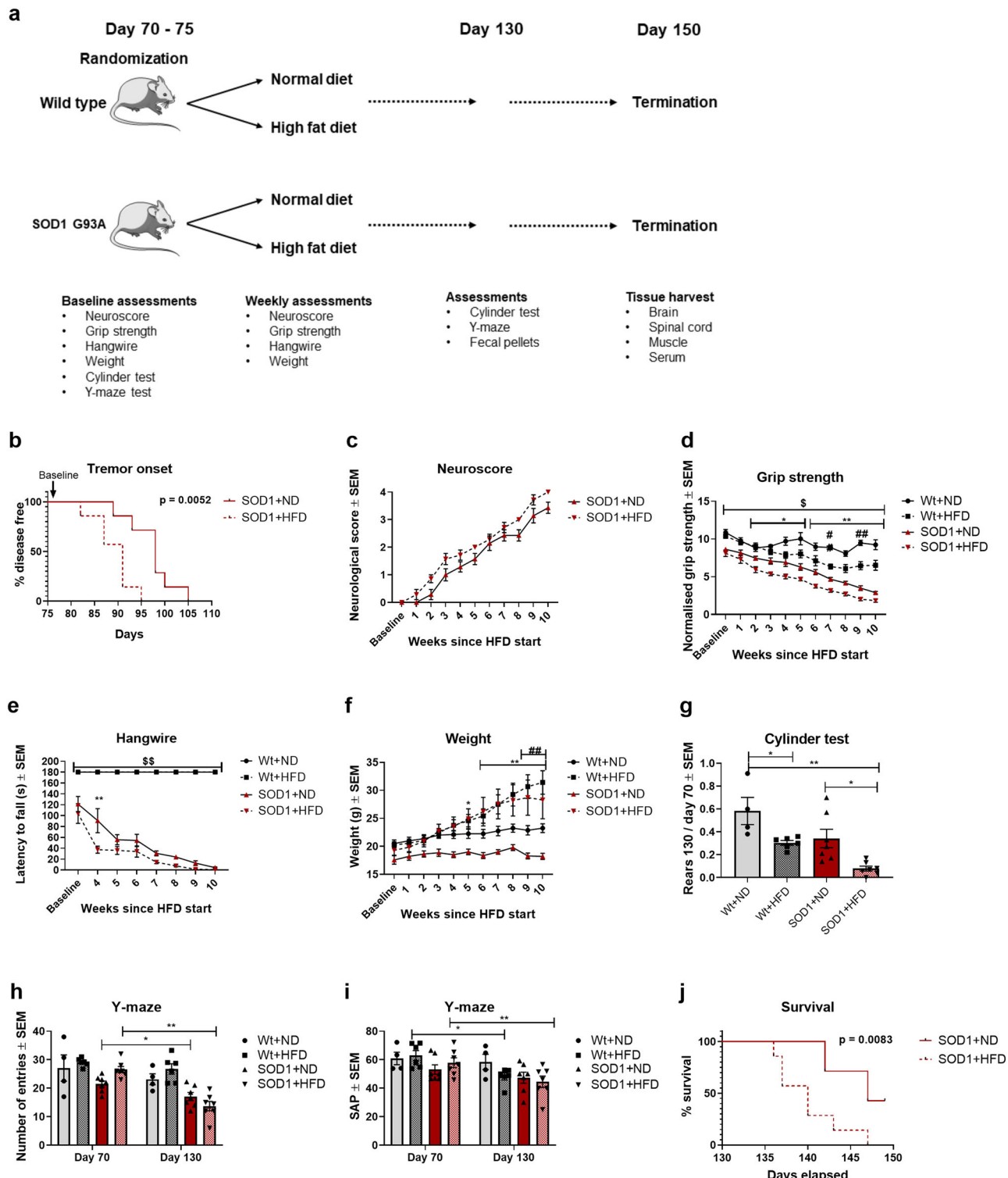

increased expression of *Cpt1a* (Fig. 9k) and *Cd68* (Fig. 9l) but no change in *Nox2* (Fig. 9m) in the spinal cord compared to SOD1 + V mice. We also found that SOD1 + CORT mice had increased expression of *Cpt1b* (Fig. 9n), *Cd68* (Fig. 9o) and decreased expression of *Glut4* (Fig. 9p) in the tibialis anterior muscle. This indicates that CORT resulted in a shift towards β-oxidation and increased reactivity of microglia and macrophages.

In summary, CORT treatment resulted in a more aggressive disease phenotype, which is consistent with previously studies in the SOD1 G93A model[76].

**Regulation of CPT1 activity results in alternations in the gut microbiome in the SOD1 G93A mouse model.** In the recent decades, studies have reported associations between changes in the gut microbiota and neurodegenerative diseases including ALS[77,78]. The gut microbiota modulates inflammation, HPA-axis activity, metabolism and other disease processes associated with ALS[79–82]. Alternations in the gut microbiome in SOD1 G93A mice correlates with inflammation, motor neuron and muscle pathology[80]. Further, the changes in metabolism due to the SOD1 G93A mutation could also result in changes in the gut

**Fig. 7 High-fat diet accelerates disease progression in the SOD1 G93A mouse model. a** Experimental setup, mice were randomized into HFD or ND at approximately day 70 (baseline) and tested weekly until 10 weeks since baseline (day 150). **b** Time point for onset of disease defined as visible tremor in hindlimbs ($n = 7$), log-rank survival analysis. **c** Neuroscore was assessed once a week. Data are presented as means ± SEM ($n = 7$). **d** Hangwire test was performed once a week. Data are presented as mean latency to fall of the grid ± SEM ($n = 7$). **e** Grip strength was measured once a week from day 70.Data are presented as means ± SEM ($n = 4$–7). **f** Mean body weight expressed in grams ± SEM ($n = 4$–7). **g** Cylinder test was performed at day 70 and day 130. Data are presented as mean ratio rears between day 130 and baseline ± SEM ($n = 4$–7). **h** Y-maze test was conducted at day 70 and day 100. Data are presented as mean entries ± SEM ($n = 4$–7). **i** Y-maze test was conducted at day 70 and day 100. Data are presented as mean spontaneous alternation percentage ± SEM ($n = 4$–7). **j** Survival was defined as neuroscore below 4, ($n = 7$), log-rank survival analysis. If nothing else is noted, data was analyzed using repeated measure two-way ANOVA followed by Tukey post hoc test. Data are representative of two experiments. *Significant differences between SOD1 + ND and SOD1 + HFD, $significant difference between SOD1 and Wt, #significant differences between Wt + ND and Wt + HFD in behavioral tests. *Significant differences between groups in serum analyses. *$p \leq 0.05$; **$p \leq 0.01$; #$p \leq 0.05$; ##$p \leq 0.01$, $$p \leq 0.05$, $$$p \leq 0.01$. Wt = wildtype, SOD1 = SOD1 G93A genotype, ND = normal diet, HFD = High-fat diet. Mouse figure used in experimental setup figures were obtained from; https://smart.servier.com/ and used according to the Creative Commons Attribution 3.0 Unsupported License.

microbiome due to the presence of metabolites in the body, which benefit some and hamper other bacteria. In addition, etomoxir and CPT1A lipid metabolism has previously been shown to modulate bacteria and the gut microbiome[24]. Based on this, we hypothesized that downregulation of CPT1 and CPT1A activity would change the gut microbiome towards a disease ameliorating composition and a disease aggravating composition by 60% HFD or CORT.

First, we examined whether SOD1 G93A mice have changes in their gut microbiota at day 70 compared to wildtype mice to assess whether disruption of the gut microbiota could play a role in disease establishment and progression. SOD1 mice presented with indications of increased intra-group microbiota diversity (α-diversity) based on Shannon, Simpson, inversed Simpson, Chao1, ACE and Pielou measures (Supplementary Fig. 5a, b). In addition, we evaluated inter-group microbiota diversity (β-diversity) based on principal component analysis (PCA). The PCA indicated that SOD1 and Wt mice had different microbial fecal compositions (Supplementary Fig. 6a). Following the evaluation of α- and β-diversity, we assessed the most relative abundant bacterial communities at phyla (Supplementary Fig. 7a), family (Supplementary Fig. 8a) and genus level using heatmap (Fig. 10a). Further, we performed differential abundance testing using DESeq2 with Benjamin–Hochberg adjustment. The differential abundance testing found multiple significant differences based on observed operational taxonomic units (OTU) (Supplementary Data 1a). This indicated that SOD1 G93A mice have changes in their microbiome before visible disease onset in accordance with the literature[80].

Based on this, we compared the fecal gut microbiome from SOD1 G93A mice at day 70 with fecal samples obtained from SOD1 G93A mice at day 130 and healthy Wt mice. SOD1 mice at day 130 had decreased α-diversity compared to SOD1 mice at day 70 (Supplementary Fig. 5c–d). In addition, PCA analysis indicated inter-group differences in the composition of the fecal gut microbiome between the groups (Supplementary Fig. 6b). Further, we assessed the most relative abundant bacterial communities at phyla (Supplementary Fig. 7b), family (Supplementary Fig. 8b) and genus level using heatmap (Fig. 10b). This indicates that the fecal gut microbiome change as the disease progresses. Following the evaluation of the most abundant communities, we performed differential abundance testing using DESeq2 with the Benjamin–Hochberg adjustment to evaluate differences between SOD1 G93A mice at day 70 compared to day 130. The differential abundance testing found multiple significant differences between the fecal gut microbiome in the SOD1 G93A mice at day 70 compared to day 130 (Supplementary Data 1b). Overall, the SOD1 day 130 group had a significant increase in communities associated with neurodegenerative diseases (Supplementary Data 1b)[80,83].

Following the findings that the gut microbiome plays a role in the establishment- and progression of disease in SOD1 G93A mice. We evaluated whether etomoxir affected the composition of the fecal gut microbiome in SOD1 G93A mice at day 130 following 60 days of treatment. First, we evaluated the α-diversity (Supplementary Fig. 5e, f) and found that both SOD1 groups had decreased α-diversity based on Chao1 and ACE measures. We also found indication of differences in β-diversity between the SOD1 + E and SOD1 + P group at day 130 (Supplementary Fig. 6c). Following the different clustering on the PCA plot, we assessed the highest relative abundant bacterial communities at phyla (Supplementary Fig. 7c), family (Supplementary Fig. 8c) and genus level (Fig. 10c) using heatmaps. Following the evaluation of the most abundant communities, we also performed differential abundance testing using DESeq2 with the Benjamin–Hochberg adjustment to evaluate differences between SOD1 + E and SOD1 + P mice. The differential abundance testing found multiple significant differences between the two treatment groups (Supplementary Data 1c). In general, the SOD1 + E group had a significant decrease in *Rikenellaceae, Akkermansiaceae, Gastranaerophilales*-, and *ϒ-proteobacteria* compared to the SOD1 G93A + P group. An increase in these communities has been associated with neurodegenerative diseases, inflammation in the intestine[84], increased lipid metabolism and degradation of the mucin layer[85]. This indicated that downregulation of CPT1A activity affected the composition of the gut microbiome, which could explain some of the differences in metabolic and inflammatory markers in the SOD1 + E mice compared with SOD1 + P mice (Fig. 2).

After we established that CPT1 lipid metabolism modulated the gut microbiome in SOD1 G93A mice, we investigated whether SOD1$^{Cpt1a/Cpt1a}$ mice had changes in their microbiome compared to SOD1 G93A mice at day 130, which could account for some of the differences seen in metabolic and inflammatory markers (Fig. 6). First, we assessed α-diversity and found that the SOD1$^{Cpt1a/Cpt1a}$ group had significantly increased α-diversity compared to SOD1 G93A mice (Supplementary Fig. 5g, h). Next, β-diversity was evaluated using PCoA plot. SOD1$^{Cpt1a/Cpt1a}$ and SOD1 G93A mice clustered into two separate groups indicating differences inter-group gut microbiome diversity (Supplementary Fig. 6d). Based on the PCoA plot we evaluated the most abundant bacterial communities at phyla (Supplementary Fig. 7d), family (Supplementary Fig. 8d) and genus level (Fig. 10d) using heatmap and found indications of differences. Following the evaluation of the most abundant communities, we performed differential abundance testing using DESeq2 with the Benjamin–Hochberg adjustment. The differential abundance testing found multiple significant differences between the groups Overall, the most remarkable changes were seen by the significant increase in *Lachnospiraceae* and *Odoribacter* (Supplementary Data 1d).

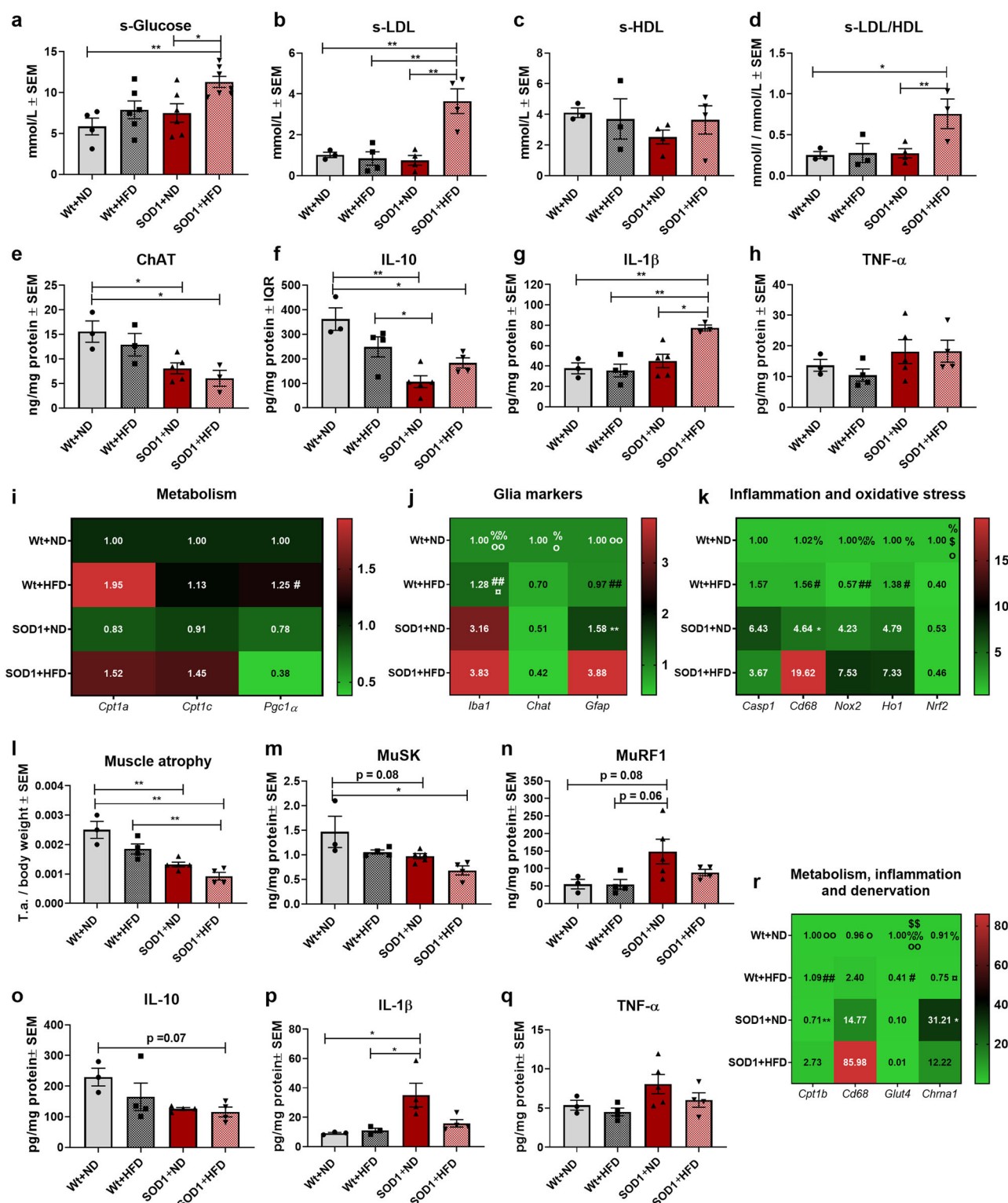

Interestingly, *Odoribacter* has been associated with resistance to HFD induction, healthy fasting serum lipid profile (high HDL and low LDL), polarization of macrophages towards the anti-inflammatory M2 type and decreased levels of TNF-α[86]. This indicated that the decreased CPT1A activity resulted in a shift in the gut microbiota in the SOD1 G93A mouse model, which potentially further could affect metabolism and inflammation. Overall, the results from etomoxir and *Cpt1a* P479L mutation experiments indicated that downregulation of CPT1 activity

resulted in a shift in the gut microbiome, which could affect the attenuation of some disease mechanisms.

Following these findings, we hypothesized that the upregulation of CPT1 activity by 60% HFD would affect the composition of the fecal gut microbiome and thereby potentially affect disease mechanisms such as inflammation. HFD resulted in a decreased α-diversity (Supplementary Fig. 5i, j) and a shift in β-diversity (Supplementary Fig. 6e). Next, we evaluated the most abundant bacterial communities at phyla (Supplementary Fig. 7e), family

**Fig. 8 60% HFD results in exacerbation of disease mechanisms in the SOD1 G93A mouse model. a** Serum glucose levels in serum samples obtained at termination. Mean mmol/L ± SEM. **b** Serum LDL levels in serum samples obtained at termination. Mean mmol/L ± SEM. **c** Serum HDL levels in serum samples obtained at termination. Mean mmol/L ± SEM. **d** Serum HDL levels in serum samples obtained at termination. Mean mmol/L ± SEM. **e** ChAT levels in lumbar spinal cord tissue homogenate obtained at termination. Mean ng/mg total protein ± SEM. **f** IL-10 levels in lumbar spinal cord tissue homogenate obtained at termination. Mean pg/mg total protein ± SEM. **g** IL-1β levels in lumbar spinal cord tissue homogenate obtained at termination. Mean pg/mg total protein ± SEM. **h** TNF-α levels in lumbar spinal cord tissue homogenate obtained at termination. Mean pg/mg total protein ± SEM. **i–k** Fold-change gene expression of metabolic, glial, inflammatory and oxidative stress genes in lumbar spinal cord tissue obtained at termination. Mean normalized fold-change gene expression ± SEM. **l** Weight of tibialis anterior muscle obtained at termination. Weight of tissue was normalized to body weight and expressed as mean ± SEM. **m** MuSK levels in tibialis anterior tissue homogenate obtained at termination. Mean ng/mg total protein ± SEM. **n** MuRF1 levels in tibialis anterior tissue homogenate obtained at termination. Mean ng/mg total protein ± SEM. **o** IL-10 levels in tibialis anterior tissue homogenate obtained at termination. Mean pg/mg total protein ± SEM. **p** IL-1β levels in tibialis anterior tissue homogenate obtained at termination. Mean pg/mg total protein ± SEM. **q** TNF-α levels in tibialis anterior tissue homogenate obtained at termination. Mean pg/mg total protein ± SEM. **r** Fold-change gene expression of metabolic, inflammatory and denervation genes in tibialis anterior tissue obtained at termination. Mean normalized fold-change gene expression ± SEM. All data was analyzed using two-way ANOVA followed by Tukey post hoc test. $N = 4$–7 for serum analysis and 3–5 for all other experiments. Data are representative of one experiment. Gene expression was normalized to β-actin and Gapdh. *Significant differences between groups in all analyses except gene expression experiments. *$p ≤ 0.05$; **$p ≤ 0.01$. Significant annotations in gene expression experiments (one = $p ≤ 0.05$, two = $p ≤ 0.01$), # = Wt+HFD vs. SOD1 + HFD, % = Wt+ND vs. SOD1 + HFD, o = Wt+ND vs. SOD1 + ND, ¤ = Wt+HFD vs. SOD1 + ND, $ = Wt+ND vs. Wt +HFD, * = SOD1 + ND vs. SOD1 + HFD Wt = wildtype, SOD1 = SOD1 G93A genotype, ND = normal diet, HFD = High-fat diet, LDL = low-density lipoproteins, HDL = High-density lipoproteins, ChAT = Choline o-acetyltransferase, MuSK = Muscle skeletal receptor tyrosine-protein kinase, MuRF1 = Muscle RING-finger protein-1.

(Supplementary Fig. 8e) and genus level (Fig. 10e) using heatmap and found multiple differences between the groups including an increase in *Lactobacillus*, *Streptococcaceae*, *Desulfovibrionaceae* and a decrease in *Lachnospiraceae*. In addition, we performed differential abundance testing using DESeq2 with the Benjamin–Hochberg adjustment. The differential abundance testing found multiple significant differences between the groups (Supplementary Data 1e). Increased levels of species of *Lactobacillus*[87] and *Streptococcaceae*[88] are associated with increased intestinal inflammation and oxidative stress, which could affect both the periphery and CNS and thereby play a role in the exacerbated disease progression in the SOD1 + HFD mice (Figs. 7, 8).

Finally, we evaluated whether CORT affected the composition of the fecal gut microbiome in SOD1 G93A mice and thereby possibly disease mechanisms at day 130, 30 days after CORT treatment was initiated. The α-diversity was increased in the SOD1 + CORT mice compared to SOD1 mice (Supplementary Fig. 5k, l). CORT administration resulted in changes in β-diversity in the SOD1 and Wt mice (Supplementary Fig. 6f). Next, we evaluated the most abundant bacterial communities at phyla (Supplementary Fig. 7f), family (Supplementary Fig. 8f) and genus level using heatmap and found multiple differences between the groups including an increase in *Lactobacillus* in both SOD1 + CORT and Wt+CORT mice (Fig. 10f). The differential abundance testing found multiple significant differences between the groups (Supplementary Data 1f). Increased levels of species of *Lactobacillus*[87] are associated with increased inflammation and oxidative stress, which could play a role in the exacerbated disease progression in the SOD1 + CORT mice (Fig. 7, 8).

In summary, we found that SOD1 G93A mice present with pathological changes in the gut microbiome, which develops as the disease progresses. The downregulation of CPT1 activity results in modulation of the gut microbiome, which could be relevant in delaying the disease progression together with target engagement in the periphery and CNS. Finally, we found that HFD and CORT results in a further disruption of the gut microbiome that could result in exacerbation of disease process such as inflammation.

## Discussion
CNS diseases represents an increasing health concern based on the aging world population. Even though several genetic risk

factors have been identified for ALS, environmental and gene-environment interaction probably account for most of the ALS cases. In this study, we provide evidence that dysregulated metabolism, and specifically CPT1 regulated lipid metabolism plays a central role in the SOD1 G93A mouse model of ALS. We show that pharmaceutical downregulation of CPT1 results in a slower disease progression when applied in the early phase of the disease, and that it has effects on several aspects of the disease phenotype, including motor and non-motor behavior. Furthermore, we provide data indicating effects if the treatment is initiated at a more progressed disease phase. In addition, the SOD1 G93A$^{Wt/Cpt1a}$ and SOD1 G93A$^{Cpt1a/Cpt1a}$ behavior data indicate that the slower disease progression is because of the diminished CPT1A activity, underpinning the potential of targeting dysregulated CPT1 lipid metabolism. These data underpin the effect seen in the Inuit population, which have the same CPT1A mutation as present in the mouse strain and show a reduced incidence of ALS[21]. In accordance with this, we have previously shown that *Cpt1a* P479L mice are resistant to EAE induction[20], and that etomoxir results in diminished inflammation[23] and autoantibody-antigen activity in EAE rats[22]. Previously, it was accepted that the CNS only utilized glucose as a substrate for energy production, but within recent years, it has become evident that the CNS can metabolize lipids to produce energy[89,90]. However, increased β-oxidation can result in increased production of reactive oxygen species and mitochondrial dysfunction and thereby disruption of homeostasis in the CNS[14]. Here, we report that downregulation of CPT1A and CPT1B by etomoxir- and CPT1A by *Cpt1a* P479L mutation impact several disease mechanisms, including inflammation, myelination, glucose metabolism and oxidative stress. Accordingly, downregulation of CPT1 by etomoxir results in amelioration of autoreactive T-cells[13], antigen presentation by dendritic cells[91] and diminished autoreactive antibodies[22], which is in accordance with our findings indicating lower levels of pro-inflammatory cytokines. In addition, we found upregulation of *Apoe* gene expression following etomoxir treatment and *Cpt1a* P479L mutation. Decreased *Apoe* is associated with increased inflammation and disrupted blood brain barrier (BBB)[22] pointing towards that targeting CPT1 results in modulation of multiple disease mechanisms. Etomoxir treated SOD1 mice had diminished labeling of CPT1A in the spinal cord, illustrating that etomoxir downregulates CPT1A, consistent with previous

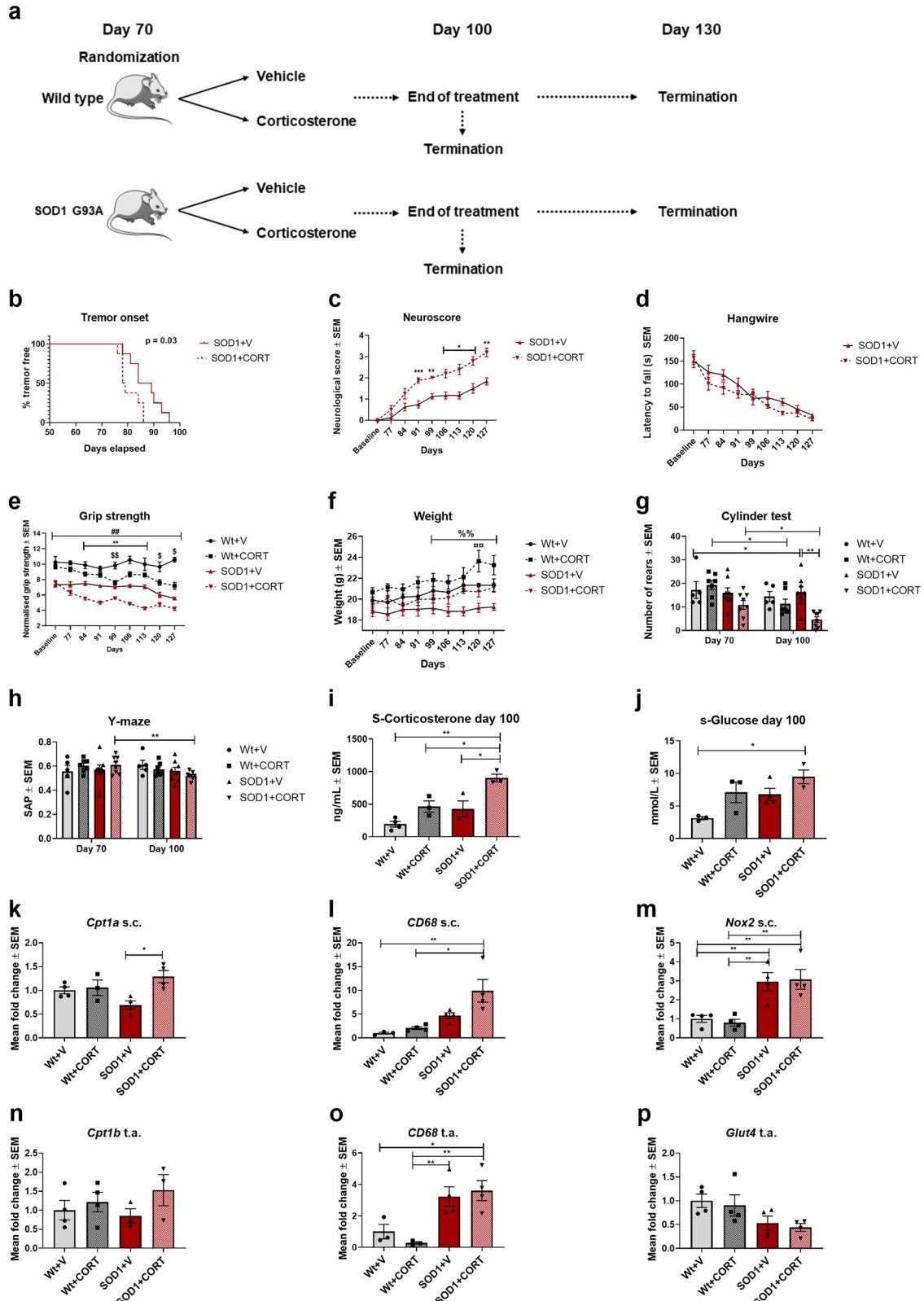

findings in vivo[23]. The beneficial effects could be due to upregulation of the glucose metabolism[92]. However, further experiments are needed to elucidate this.

A major problem in the treatment of neurodegenerative diseases is the development of drugs that can cross the BBB. Etomoxir is indicated to affect the metabolism both within the CNS[93] and in the periphery[25]. Data presented in the etomoxir experiments indicate that target engagement takes place both within the CNS and periphery regarding, e.g., inflammation. However, another drug that downregulates β-oxidation is ranolazine, which has shown beneficial effects by downregulating CPT1B in the muscles in SOD1 G93A mice[50] and inflammation in vitro[94]. Therefore, we *hypothesize* that peripheral target engagement in SOD1 G93A mice could prove to have effects by

**Fig. 9 Corticosterone accelerates disease progression in the SOD1 G93A mouse model. a** Experimental setup, mice were randomized into treatments at approximately day 70 and tested weekly. 3–4 mice were terminated at day 100 and the rest (3–4) were terminated at day 130. **b** Time point for onset of disease defined as visible tremor in hindlimbs ($n = 8$), log-rank survival analysis. **c** Neuroscore was assessed once a week. Data are presented as means ± SEM ($n = 8$). Mixed effect analysis with Tukey post hoc test. **d** Hangwire test was performed once a week. Data are presented as mean latency to fall of the grid ± SEM ($n = 8$). Mixed effect analysis with Tukey post hoc test. **e** Grip strength was measured once a week from day 70. Data are presented as means ± SEM ($n = 5$–8). Mixed effect analysis with Tukey post hoc test. **f** Mean body weight expressed in grams ± SEM ($n = 5$–8). Mixed effect analysis with Tukey post hoc test. **g** Cylinder test was performed at day 70 and day 100. Data are presented as mean rears ± SEM ($n = 5$–8). Repeated measure two-way ANOVA with Tukey post hoc test. **h** Y-maze test was conducted at day 70 and day 100. Data are presented as mean spontaneous alternation percentage ± SEM ($n = 5$–8). Paired $t$-test. **i** Corticosterone levels in serum obtained at day 100 was analyzed using ELISA. Data are presented as mean ng/mL ($n = 3$–4). Two-way ANOVA with Tukey post hoc test. **j** Glucose levels in serum at day 100. Data are expressed as mean mmol/L ± SEM ($n = 3$–4). Two-way ANOVA with Tukey post hoc test. **k–m** Cpt1a, Cd68, and Nox2 gene expression in the lumbar spinal cord from mice terminated at day 130. Data are presented at mean normalized fold-change. Expression was normalized to β-actin and Gapdh. N = 3–4. Two-way ANOVA with Tukey post hoc test. **n–p** Cpt1b, Cd68, and Glut4 gene expression in the tibialis anterior muscle from mice terminated at day 130. Data are presented at mean normalized fold-change. Expression was normalized to β-actin and Gapdh. N = 3–4. Two-way ANOVA with Tukey post hoc test. Data are representative of two animal experiments. Serum and RT-qPCR experiment was conducted once. *Significant differences between SOD1 + V and SOD1 + CORT, #significant difference between SOD1 and Wt, $significant differences between Wt + V and Wt + CORT, % significant differences between SOD1 + V and Wt-CORT in body weight analysis, ¤significant difference between Wt+CORT and SOD1 + CORT in body weight analysis. *$p \leq 0.05$; **$p \leq 0.01$; #$p \leq 0.05$; ##$p \leq 0.01$, $$p \leq 0.05$, $$$p \leq 0.01$, % $p \leq 0.05$, %%$p \leq 0.01$. WT = wildtype, SOD1 = SOD1 G93A genotype, V = Vehicle, CORT = Corticosterone. Mouse figure used in experimental setup figures were obtained from; https://smart.servier.com/ and used according to the Creative Commons Attribution 3.0 Unsupported License.

amelioration of peripheral hypermetabolism and thereby, e.g., inflammation and oxidative stress.

In this study, we hypothesized that upregulation of CPT1 by 60% HFD (based on saturated animal fats) would result in a more severe disease phenotype. We found that 60% HFD resulted in a more severe disease phenotype, possibly by increased inflammatory activity and oxidative stress. The effect of diets on disease progression in ALS in vivo models have been evaluated previously[61]. However, the majority of studies are incomparable because of differences in genetic background- and the composition of the HFD. The studies indicating a positive effect are characterized by diets containing high amount of unsaturated omega-3 fatty acids[61], which are characterized by anti-inflammatory effects[15]. Nonetheless, studies in animal models of other neurodegenerative diseases, e.g., multiple sclerosis[20] and Parkinson's-[95] and Alzheimer's disease[96] have shown that HFD results in more severe disease progression, demyelination, inflammation, death of neurons and brain atrophy. HFD results in upregulation of CPT1and thereby a shift towards lipid metabolism[57]. A cross-sectional study found that intake of food with a traditional high content of saturated fat correlated with a lower functional ALSFR-R score[60]. A double-blinded placebo-controlled randomized clinical phase 2 trial found that ALS patients fed a diet with high fat had more adverse events and deaths compared to patients receiving a diet with high levels of carbohydrates[62]. This indicates that HFD might be undesirable, but larger clinical trials, and standardized in vivo studies are needed. We also hypothesized that CORT would exacerbate disease progression in SOD1 mice because cortisol is increased in several CNS diseases including ALS[73,74] and upregulates β-oxidation[70]. In accordance with this, we show that CORT administration for 30 days results in a more severe disease progression in SOD1 G93A mice based on behavior data. In addition, our data indicates that CORT resulted in increased levels of inflammatory cells expressing Cd68. This is in agreement with a previous study that found a negative correlation between corticosterone levels and disease progression in SOD1 G93A mice following chronic restraint stress[76]. In addition, chronic upregulation of the HPA-axis results in increased inflammation[76]. Nonetheless, further investigations of the effects of CORT in SOD1 mice are needed.

In the last few years, the role of the gut microbiota in ALS and in the SOD1 G93A mouse model has received increasing focus[78,80,97]. The gut microbiome correlates with multiple disease mechanisms in SOD1 G93A mice, including inflammation, muscle pathogenesis and motor neuron pathology[80]. Accordingly,

the intestinal barrier is disrupted in SOD1 G93A mice resulting in increased levels gut metabolites and inflammatory cytokines in the blood, which further can affect both peripheral tissues and the CNS. Based on this, we hypothesized that downregulation of CPT1 activity might result in alternations in the gut microbiome directly and by the change in metabolite levels due to the downregulated β-oxidation, thereby affecting disease mechanism in SOD1 G93A mice. We found that downregulation of CPT1 by etomoxir and CPT1A activity by Cpt1a P479L mutation resulted in alternations in the fecal gut microbiome including a decrease in microbiome communities associated with disease. In addition, the downregulated CPT1A activity resulted in an increase of Lachnospiraceae and Odoribacter, which are associated with decreased muscle atrophy, inflammation and possibly upregulated glucose metabolism[80,86]. This point towards that CPT1A activity directly affects the fecal gut microbiome composition[24]. In addition, we hypothesized that 60% HFD and CORT would affect the fecal gut microbiome directly but also indirectly by the upregulation of CPT1. HFD and CORT resulted in increased levels of communities such as Lactobacillus, Streptococcaceae, Desulfovibrionaceae and a decrease in some OTUs of Lachnospiraceae. These increased communities are associated with increased production of inflammatory cytokines within the gut and potentially insulin resistance due to obesity[87,88]. This indicates that directly and indirectly modulation of the gut microbiota by environmental factors such as diet and stressors could play a role in the disruption of metabolism and thereby affect multiple mechanism such as inflammation and metabolism.

In summary, we report that CPT1 lipid metabolism potentially plays a role in the modulation of disease activity both in the periphery and within the CNS in the SOD1 G93A mouse model possibly by regulation of glucose metabolism. This adds to the increasing knowledge on the interaction between genetic susceptibility and environment in the development of ALS, and specifically dysregulated metabolism being a central pathway and possibly a target for disease modification. Nonetheless, more mechanistic studies are warranted.

## Methods

**Animals.** All experiments were approved by the Danish Animal Experiment Inspectorate (2017-15-0202-00088) and followed the National and European guidelines for conducting animal experiments. Animal experiments was conducted according to the ARRIVE guidelines. Mice were housed in IVC cages in a high

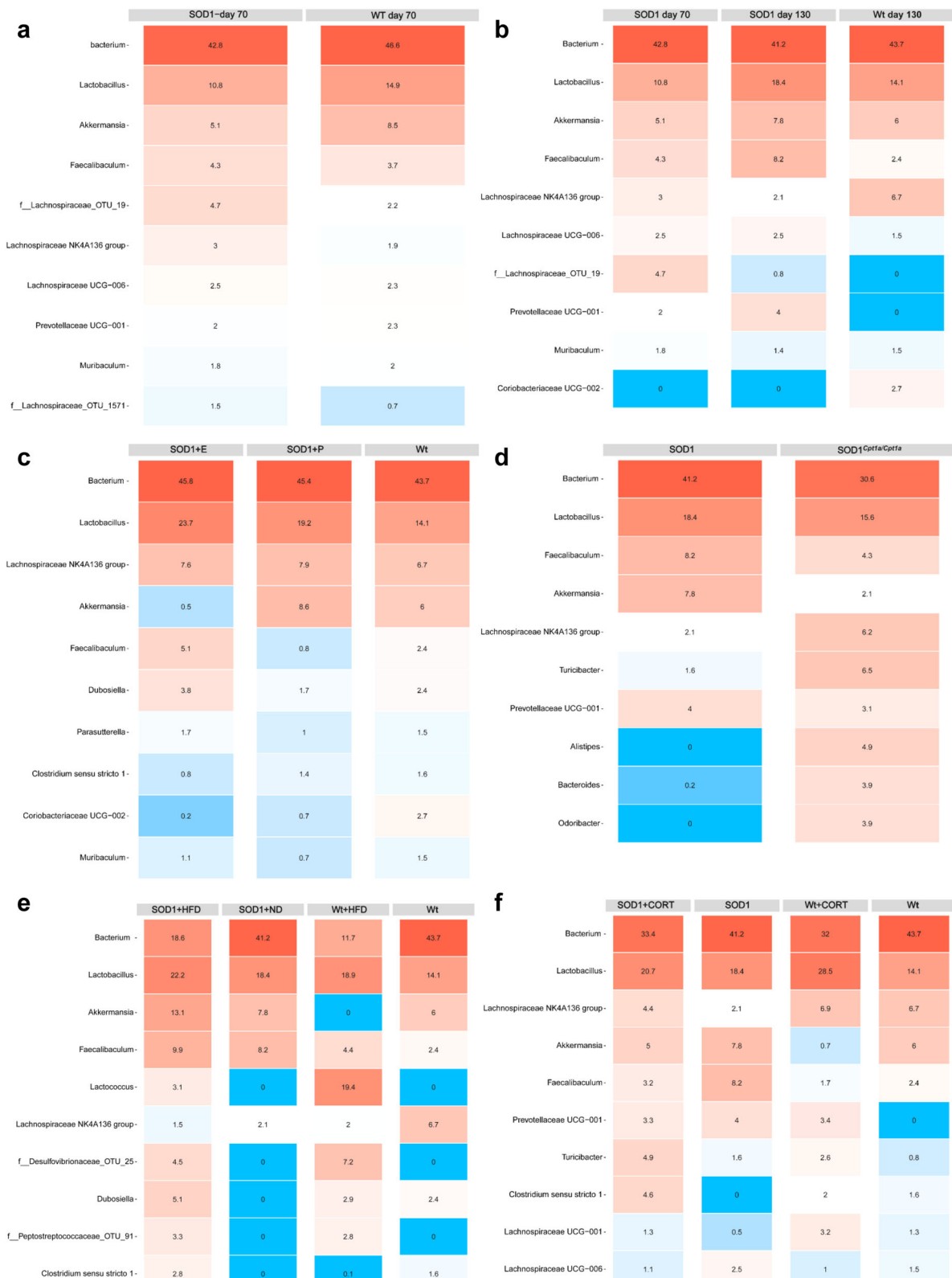

barrier facility at Aalborg University with a room temperature of 21 °C. The mice were kept at a 12-hour light/dark cycle and had ad libitum access to food and water. Sample size was estimated based on previous studies and a pilot study described in the manuscript.

**SOD1 G93A mouse model**. B6.Cg-Tg(SOD1*G93A)1Gur/J mice (Stock 004435) (SOD1) were purchased from Jackson Laboratory (Bar Harbor, USA). The congenic SOD1 mice were maintained in our animal facility by crossing hemizygote

SOD1 male mice with female C57Bl/6J mice. Litters were genotyped according to established protocol using DNA extracted from ear tissue punches[80]. Male SOD1 mice were used to maintain the colony, and SOD1 female and their wildtype littermates were used for experiments. All animals were assessed for human endpoints daily, weight twice a week, and neurological score once a week. SOD1 males ($n = 9$) and females ($n = 5$) were evaluated by neurological score, grip strength, hangwire test and y-maze test at day 140 to evaluate differences between the males and females as described under clinical behavioral tests. In addition, survival was

**Fig. 10 Modulation if CPT1 activity by etomoxir, genetic inhibition, high-fat diet, and corticosterone modulates the gut microbiota in SOD1 G93A mice.**
**a** Heatmap illustrating the ten most abundant communities at the genus level in fecal samples from SOD1 mice and Wt mice at day 70. Values represents mean relative abundancy. **b** Heatmap illustrating the ten most abundant communities at the genus level in fecal samples from SOD1 mice at day 70, day 130, and Wt mice at day 130. Values represents mean relative abundancy. **c** Heatmap illustrating the ten most abundant communities at the genus level in fecal samples from SOD1 + P, SOD1 + E, and Wt mice at day 130. Values represents mean relative abundancy. **d** Heatmap illustrating the ten most abundant communities at the genus level in fecal samples from SOD1 and SOD1$^{Cpt1a/Cpt1a}$ mice at day 130. Values represents mean relative abundancy. **e** Heatmap illustrating the ten most abundant communities at the genus level in fecal samples from SOD1 + ND, SOD1 + HFD, Wt+ND, and Wt+HFD mice at day 130. Values represents mean relative abundancy. **f** Heatmap illustrating the ten most abundant communities at the genus level in fecal samples from SOD1, SOD1 + CORT, Wt, and Wt+CORT mice at day 130. Values represents mean relative abundancy. Fecal pellet samples were harvested at day 70 or day 130, $n = 4$–5. Data are representative of one 16S rRNA sequencing experiment. Wt = wildtype, SOD1 = SOD1 G93A genotype, SOD1$^{Cpt1a/Cpt1a}$ = SOD1 G93A mice with homozygote Cpt1a p479l mutation. E = etomoxir, P = placebo, HFD = high-fat diet, CORT = corticosterone.

evaluated as described under neurological score, onset of disease and survival analysis.

**SOD1 G93A$^{Wt/Cpt1a}$ and SOD1 G93A$^{Cpt1a/Cpt1a}$ mouse models.** B6.Cg-Tg (SOD1*G93A)1Gur/J male mice were crossed with female B6.Cpt1a P479L homozygote mice (strain generated in collaboration with the Netherlands Cancer Institute, Amsterdam, as previously published[20] with the use of the genomic Cpt1a promoter, causing no difference in expression in different tissues) to obtain SOD1 mice with a heterozygote Cpt1a P479L mutation (SOD1 G93A$^{Wt/Cpt1a}$)[24]. SOD1 G93A$^{Wt/Cpt1a}$ mice were crossed with female Cpt1a P479L homozygote mice to obtain SOD1 G93A$^{Cpt1a/Cpt1a}$. All mice were genotyped using ear punch tissue as previously described[20,80]. All animals were assessed for human endpoints daily, weighted twice a week, and neurological score once a week.

**CPT1A antagonist studies.** A pilot study was conducted using female SOD1 ($n = 10$) and wildtype ($n = 4$) littermates. SOD1 mice were randomized into treatment with the irreversible CPT1A antagonist, etomoxir (5 mg/kg) ($n = 7$), or placebo (olive oil) ($n = 3$) from day 100. Treatment was administered one a daily basis by oral gavage. Mice were evaluated weekly by neurological score, weight, and hangwire test as described under clinical behavior test. Following the pilot study, a larger study was setup. Female SOD1 ($n = 18$) and wildtype ($n = 9$) littermates were randomized into treatment with etomoxir or placebo from day 70. Mice were evaluated weekly by neurological score, grip strength, hangwire test, rotarod test, and at day 70, day 100 and day 130 mice were evaluated by cylinder, and y-maze test. Moreover, fecal samples were collected at day 70 before treatment start ($n = 5$), and at day 130 for the two SOD1 treatment groups. Serum, brain, muscle, and spinal cord tissue was obtained when mice were euthanized between day 145–150. Another CPT1A antagonist study was conducted from day 70 and until day 130 to obtain spinal cord tissue for immunohistochemistry. Female SOD1 ($n = 11$) and wildtype littermates ($n = 4$) were randomized into treatment with etomoxir or placebo and evaluated by neurological score and grip strength once a week, as described below. At day 130 mice were anesthetized using isoflurane and quickly perfused with 1xPBS followed by 4% paraformaldehyde and spinal cords were harvested for further analyzes.

**SOD1$^{Wt/Cpt1a}$ and SOD1$^{Cpt1a/Cpt1a}$ studies.** SOD1 ($n = 5$), SOD1$^{Wt/Cpt1a}$ ($n = 8$), and SOD1$^{Cpt1a/Cpt1a}$ ($n = 10$) female mice were evaluated weekly from day 70 by neurological score, grip strength and hangwire test. Moreover, mice were evaluated at day 70, 100, and 130 using cylinder ($n = 7$–11), and y-maze test ($n = 6$–9). Not all mice was evaluated by the cylinder and y-maze test, which explains differences in $n$. Fecal samples were obtained from SOD1$^{Cpt1a/Cpt1a}$ mice at day 130. For onset of disease (tremor) evaluation, female SOD1 ($n = 15$), SOD1$^{Wt/Cpt1a}$ ($n = 16$), and SOD1$^{Cpt1a/Cpt1a}$ ($n = 12$) mice were used. Serum, muscle, and spinal cord tissue samples were obtained when mice were euthanized between day 145–160. Another group of female SOD1$^{Wt/Cpt1a}$ ($n = 6$), SOD1$^{Cpt1a/Cpt1a}$ ($n = 3$) and SOD1 mice ($n = 4$) was evaluated by neurological score from day 70 and until day 132. In addition, male SOD1$^{Cpt1a/Cpt1a}$ ($n = 25$) and SOD1 mice ($n = 10$) was used to evaluate differences in survival.

**SOD1 G93A High-fat diet study.** Female SOD1 ($n = 14$) and their wildtype littermates ($n = 10$) were randomized into receiving normal diet (kcal% respectively: protein 29, carbohydrate 65.5, fat 5.5, Brogaarden, Denmark) or high-fat diet (kcal% respectively: protein 20, carbohydrate 20, fat 60, Brogaarden, Denmark; based on D12492 research diets, New Brunswick, USA) from day 70. Clinical symptoms were evaluated weekly by neurological score, grip strength, and hangwire test. Weight was assessed twice a week. Moreover, mice were evaluated at day 70, and day 130 using cylinder, and y-maze test. Serum, muscle, and spinal cord tissue samples were obtained when mice were euthanized between day 145–150.

**SOD1 G93A corticosterone study.** Female SOD1 ($n = 18$) and their wildtype littermates ($n = 12$) were randomized into receiving vehicle or corticosterone

(20 mg/kg) (Sigma, CAT# 27840) from day 70 and until day 100 by oral gavage as previously described[98]. Clinical symptoms were evaluated weekly by neurological score, grip strength and hangwire test. Weight was assessed twice a week. Moreover, mice were evaluated at day 70, and day 100 using cylinder and y-maze test. Serum samples were obtained when mice were euthanized at day 100 and day 130, respectively. Spinal cord and tibialis anterior muscle tissue was harvested at day 130. Fecal samples was obtained from mice at day 130 following 30 days corticosterone washout period.

**Clinical behavioral tests.** All clinical behavioral tests were performed between 9 am and 2 pm. All tests were performed in the same laboratory and mice were placed in the room one hour before test sessions to allow time for acclimatization. Test equipment was cleaned with 70% ethanol between each animal.

**Neurological score, onset of disease, and survival analysis.** Mice were evaluated by the same experimenter weekly. The experimenter was blinded to treatment group and genotype. Mice were given a neurological score between zero to five as previously described[30,31]. Zero = no tremor in hindlimbs and full extension of hindlimbs when suspended by its tail. One = tremor in hindlimbs and full extension of hindlimbs when suspended by its tail. Two = tremor in hindlimbs and unable to extend hindlimbs when suspended by its tail. Three = tremor in hindlimbs, unable to extend hindlimbs when suspended by its tail and wobbling gait. Four = tremor in hindlimbs, unable to extend hindlimbs when suspended by its tail difficulty walking with paralysis of one of both hindlimbs. Five = tremor in hindlimbs, unable to extend hindlimbs and unable to get up within 30 s when placed on its side. Onset of disease was defined as the time point when tremor in the hind legs was present as previously described[30]. Owing to ethical reasons and based on the guidelines in the Animal Facility, mice were terminated if they reached a neurological score of four or latest at day 160. Based on this, survival was defined as a neurological score below 4 at the final day of experimentation. Owing to the fact that mice had to be terminated at day 160, some mice did not reach a neurological score of 4 and, therefore, some of the groups have censored data in the survival analysis.

**Hangwire test.** Mice were gently placed on a wire grid lid and turn upside down. The latency to fall was noted. The maximum cutoff time was set to 180 s. Each mice received three trials per sessions. The highest latency to fall was used for subsequent statistics[31].

**Rotarod test.** Rotarod test (Rotamex-5 RotaRod, Columbus Instruments, Columbus, Ohio, USA) was used with an acceleration from 4 to 40 RPM over 5 min[29]. Mice were acclimatized to the rotarod over three consecutive days before the first test session. Each mouse was tested three times per test session to obtain a mean latency to fall (s).

**Grip strength test.** Grip strength was evaluated using Grip strength meter (Bioseb, France). Briefly, the mouse was placed on a wire grid at pulled by its tail. The maximum tension was measured in grams Each mouse received four trials at each session and a mean grip strength was calculated. The mean grip strength was normalized to weight as previously described[24].

**Cylinder test.** The cylinder test was used to evaluate the sensorimotor function and spontaneous activity as previously described[32]. Mice were transferred into a quiet room with low illumination and placed in the glass cylinders. Test was recorded for 3 min using a video camera. The number of rears were counted by four blinded raters.

**Y-maze test.** The y-maze test was constructed according to Maze Engineers (USA). The mice were placed in the y-maze for 5 min to freely explore the three arms. The y-maze test was recorded by video and the number of entries and triplets

were noted by four blinded raters as previously described[99]. The mean spontaneous alternation percentage was calculated as previously described[99].

**Serum glucose-, low-density lipoprotein cholesterol-, and high-density lipoprotein cholesterol assays.** Blood samples were obtained from mice by retro orbital puncture. Mice were in fed-state. Samples were placed at room temperature for 40 min and afterwards samples were centrifuged at $3500 \times g$ for 15 min and the supernatant was transferred and stored at $-80\,^{\circ}C$ until further analyses. Serum analyses of glucose- (Crystal Chem, CAT#81692), low-density lipoprotein cholesterol (LDL) (Crystal Chem, CAT#79980), high-density lipoprotein cholesterol (HDL) (Crystal Chem, CAT#79990) and corticosterone (Crystal Chem, CAT#80556) levels were performed using commercial assay kits according to the manufactures instructions. Samples- and standards were run in duplicates and were analyzed using a plate reader (PerkinElmer). Concentrations were obtained, based on standard curve methods as described in the manufacture's protocols.

**Reverse-transcriptase quantitative polymerase chain reaction (RT-qPCR).** RNA was extracted from brain, spinal cord and tibialis anterior muscle using GeneJET RNA purification kit (ThermoFisher, CAT# K0732). Following RNA extraction, quality and quantity was evaluated using NanoDrop. cDNA synthesis and qPCR was performed as previously described[20]. All samples were run in duplicates. The following primers RT² qPCR Primer Assay for Mouse were purchased from Qiagen: *Igf1* (PPM03387F-200), *Mbp* (PPM04745F), *Chrna1* (PPM03976A), *Cpt1b* (PPM57688A), *Casp1* (PPM02921E), *Cd68* (PPM03976A). The following primers were purchased from TagCopenhagen (Denmark): Cpt1a rev: GGAGGTTGTCCACGAGCCAG, fwd: TCATCAGCAACCGGCCCAAA. *Cpt1c* rev: TTTTCCAGGAGCGCAGGG, fwd: CTGACCTCTGACCGGTGGGC. *Nrf2* rev: GGGGATATCCAGGGCAAGCG), fwd: CGCCAGCTACTCCCAGG TTG. *Il-17a* rev: GTCCAGCTTTCCCTCCGCAT, fwd: CCTGGACTCTCCACC GCAATG. *Ifn-Υ* rev: CTTCCCCACCCCGAATCAGC, fwd: GCCAAGTTTG AGGTCAACAACCC. *β-actin* rev: TCGTCATCCATGGCGAACTGG, fwd: CT GTCGAGTCGCGTCCACC. *Ho1* rev: ATCCTGGGGCATGCTGTCGG, fwd: GAGCCGTCTCGAGCATAGCC. *Nox2* rev: TTCAGCCAAGGCTTCAGGGC, fwd: TGGACGGCCCAACTGGGATA. *Pgc1α* rev: TGCGGTATTCATCCC TCTTG, fwd: CACCGCAATTCTCCCTTGTA. *Hprt* rev: CGCTAATCACGA CGCTGGGA, fwd: GGGGAGAGCGTTGGGCTTACC. *Slc2a4* (Glut4) rev: CCTCCCGCCCTT AGTTG, fwd: CTGCAA AGC GTA GGT ACC AA. *Gapdh* rev: TGTGAGGGAGATGCTCAGTG, fwd: GTGGAC CTCATGGCCTACAT. *Apoe* rev: TGT GTG ACT TGG GAG CTC TG, fwd: GTGCTGTTGGTCACATTGCT. *Iba1* rev: AGTTGGCTTCTGGTGTTCTTTGTTT, fwd: GTTCCCAAGACCCA TCTAGAGCTG. *Gfap* rev: CAGGGCTCCATTTTCAATC, fwd: ACAGACT TTCTCCAACCTCCA. *Chat* rev: GGATGGGAGCAGGGTTAGTA, fwd: CACC AACAGCAAAGGAAAGA. Gene expression in spinal cord- and tibialis anterior tissue was normalized to *Gapdh* and *β–actin*. Gene expression in brain tissue (Supplementary Fig. 3) was normalized to *Hprt* and *β–actin*. Mean fold-change expression was calculated according to $2^{-\Delta\Delta Ct}$ method.

**Immunohistochemistry.** Immunohistochemical peroxidase staining on the lumbar spinal cord was performed to evaluate morphological changes between genotypes and treatment groups. Tissue was processed as previously described[20,23]. The following primary antibodies were used: MBP 1:800 (Abcam, CAT# ab7349), GFAP 1:1000 (Agilent, CAT# Z033429-02), CPT1A 1:400 (ThermoFisher Scientific, CAT# PA5-69347), IBA-1 1:500 (Wako, CAT#013-27691), CHAT 1:400 (Merck Millipore, CAT#ab144p). The following secondary HRP-conjugated antibodies were used: Goat Anti-rat 1:500 (ThermoFisher Scientific, CAT# PA1-84708), and Goat Anti-rabbit 1:500 (Dako, CAT# P0448). The sections were imaged using a Leica microscope as previously described[23]. All microscopy settings were kept identical during the entire acquisition process.

**Enzyme-linked immunosorbent assay (ELISA).** Spinal cord tissue and muscle tissue were weighted and quickly homogenized in ice cold PBS according to manufactures protocols followed by centrifugation at $4500 \times g$ for 5 min at $4\,^{\circ}C$. The supernatant was transferred and used in the subsequent ELISA experiments. The following commercial ELISA kits were used CHAT (Biosite, CAT#EXX-74AG9R-96), CX3CR1 (Biosite, CAT#EXX-FR3AVA-96), IL-10 (Invitrogen, CAT# 88-7105-22), IL-1β (Invitrogen, CAT# 88-7013-22), TNF-α (Invitrogen, CAT# 88-7324-86), MuSK (Aviva systems biology, CAT#OKEH07075), MuRF1 (Biosite, CAT#EXX-ARLE1G-96), Atrogin-1 (FBXO32) (Biosite, CAT#EKK-6W410X-96), Myogenin (Aviva systems biology, CAT#OKEH03290). Serum was obtained as described above, diluted 1:100 in commercial included buffer and used in the mouse NF-L ELISA kit (LSBio, CAT#LS-F70035). All samples were run in duplicates. All procedures were done according to the manufactures protocols. ELISAs' were read at 450 nm using a plate reader (PerkinElmer). Concentrations were obtained based on standard curve methods, as described in the manufacturers' protocols. Following, concentrations were normalized to total protein concentration in the respective samples. Total protein concentrations were estimated using Nanodrop technology.

**Fecal gut 16S rRNA microbiome analyses.** Fecal pellets were stored at minus 80 ° C until further analyzes. DNA was extracted from fecal pellets using *Quick*-DNA Fecal/Soil Microbe Miniprep Kit (Zymogene, CAT# D6010) according to the manufacture's protocol. Quantity and quality of extracted DNA was assessed using Nanodrop. Following extraction, DNA was handed over to DNASense Aps (Aalborg, Denmark) for 16s rRNA V4 region sequencing and bioinformatics processing. Library preparation-, sequencing and bioinformatics were done as previously published[24]. Heapmaps, principal component analysis and DESeq2 with Benjamin–Hochberg adjustment analysis were conducted using https://dnasense.shinyapps.io/dnasense/ provided by DNASense ApS (Aalborg, Denmark) based on the ampvis R package described in ref. [24].

**Statistics and reproducibility.** All statistics was performed using Graphpad Prism v. 8.0 except for the 16s rRNA microbiome analyzes. Normality and variance was evaluated before performing statistical analysis. Pairwise comparison were generated with two-tailed paired *t*-test. Independent groups were compared using two-tailed unpaired *t*-test. Comparison of more than two independent groups was performed by one-way ANOVA followed by Tukey multiple comparison test. Two-way ANOVA followed by Tukey multiple comparison test was performed when more than three groups were compared with two factors. Repeated measure two-way ANOVA were conducted when several groups were tested multiple times followed by post hoc Bonferroni or Tukey multiple comparison test. Survival analysis was done by log-rank test. Outliers were evaluated by Grubbs method. *P*-values, *n*-values, definition of center and dispersion measurements are indicated in the associated figure legends for each figure. Number of repeated experiments are described in the figure legends. Repeated experiments produced comparable results and findings were considered reproducible.

**Reporting summary.** Further information on research design is available in the Nature Research Reporting Summary linked to this article.

## Data availability
Source data for the microbiome studies are freely available from Mendeley Data using https://doi.org/10.17632/6zt29bjbzd.1. Source data for figures are available in Supplementary Data 2. In addition, data that support the findings of this study are available from the corresponding author (J.D.V.N) upon reasonable request.

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

## Acknowledgements

We acknowledge Aage and Johanne Louis-Hansen Foundation (17-2B-1297), Gangsted Foundation (A33337), A.P. Møller Lægefonden (18-L-0060) and Foundation for Neurological Research (R61-A2933) for providing funding for experiments and Ph.D.-fellowship for M.S.T. We acknowledge Meta-IQ Aps (Aarhus, Denmark) for providing etomoxir. Finally. We acknowledge DNASense Aps for performing 16s rRNA sequencing.

## Author contributions

M.S.T. established the mice models in the laboratory, planned and conducted the in vivo experiments, serum analyses, RT-qPCR, preparation of fecal samples for 16s rRNA sequencing, performed data analysis and wrote the manuscript. D.C.A. and P.H. assisted during in vivo experiments. P.H., K.M., N.W., and U.B.K. assisted in blinded rating of clinical videos. K.E.O. performed histology experiments. L.B. assisted during genotyping and RT-qPCR on spinal cords- and muscles. J.L.H. and A.E.B. performed RT-qPCR experiments on brains. L.J.K., C.E.J.P., and I.J.H. developed the *Cpt1a* P479L mouse model. J.D.V.N. supervised development of experimental setups, and revised the manuscript. All authors reviewed the manuscript.

## Competing interests

The authors declare no competing interests.
