## [Peer Review File · Communications Biology]

Reviewers' Comments:

Reviewer #1:

Remarks to the Author:

Trabjerg et al. report that modulating carnitine palmitoyl transferase 1 (CPT1) activity effects disease progression in SOD1G93A mice, a familial mouse model of amyotrophic lateral sclerosis (ALS). CPT1 is an enzyme that facilitates the transport of medium-long chained fatty acids across the outer mitochondrial membrane to enable β -oxidation. There are 3 different isoforms of CPT1, each with varying levels of expression in different tissue types. CPT1A is expressed ubiquitously, CPT1B is limited to muscle and adipose tissue whereas CPT1C is found in the CNS. The authors show that either pharmacological inhibition of CPT1 activity with Etomoxir or genetic knockdown of CPT1A ameliorates disease symptoms in SOD1G93A mice. Additional experiments show that disease progression is accelerated in SOD1G93A mice following consumption of a 60% high fat diet or treatment with corticosterone. Finally, the authors show that downregulating CPT1 shifts the gut microbiota communities in SOD1G93A mice. Collectively, these findings suggest that CPT1 mediated lipid metabolism plays an important role in regulating disease progression in SOD1G93A mice and suggests modulating CPT1 might be a therapeutic target for ALS.

Overall, I believe the results of these studies are valid, provide novel insights into the pathogenesis of ALS, and are of interest to scientists from different disciplines. However, methodological details of the experiments need to be addressed prior to being considered for publication (see below). The appropriate statistical tests were used for the pharmacological, genetic, HFD and corticosterone data. However, I am not in position to comment if the correct analyses were used for the microbiome data. Previous literature was cited correctly. Furthermore, the clarity and flow of the manuscript needs to be significantly improved especially for readers that are less familiar with lipid metabolism. Please see suggestions below.

Concerns

The title is somewhat subjective – get rid of strongly. Based on the results I would recommend changing the title to “Downregulating carnitine palmitoyl transferase 1 slows disease progression in the SOD1 G93A mouse model”.

The methods used to calculate the Neuroscore and survival needs to be explicitly stated in the methods. Interpretation of data presented in figures 1c, 2c, 2g, 3c and 4c is not possible without these details. I did not have access to the papers published referenced for these methods (which may also be true for potential readership).

For Figures 1b, 2b, 3b and 4b the median age of tremor onset for each group needs to be reported along with the p values.

The survival data present in Figure 2G is incomplete. At what age were 0% of the genetically manipulated mice alive? The median survival ages for each group also need to be stated in the results.

In the figure legends N values are given, but then it is stated the data represents 2-3 different animal studies (depending on the study). It is unclear if the N value represents the total number of mice used across all of the experiments or if it represents the N value per experiment. For example, for Figure #1, is the total N value 4-10 per group or 12-30 per group? Please clarify.

The target enzymes (CPT1A, B and C) and tissues (peripheral vs. CNS) of Etomoxir are unclear. Please edit the manuscript to include which isoforms of CPT1 are inhibited by Etomoxir and if it is known if the compound crosses the blood brain barrier. Similarly, for the genetic studies, the promoter used to manipulate CPT1A levels should be stated. We need to know if CPT1A expression and enzyme activity is altered in both peripheral and CNS tissues.

The relative importance of CPT1 target engagement in peripheral vs CNS tissues should be included in the discussion. Making CNS penetrant compounds is difficult and if your data suggests this may not be necessary, this would be important to know for developing CPT1 inhibitors as potential therapeutics.

Experiments 3 and 4 complement the first 2 studies nicely and are in agreement with some of the previously published work in the same model. Presumably, experiments 3 (HFD study) and 4 (corticosterone study) were carried out to show that promoting acyl-coa accumulation, increasing the rate of β -oxidation, and impairing glucose metabolism have the opposite effect of down-regulating CPT1 enzyme activity (experiments 1 and 2). This connection between experiments may be challenging for individuals that have a more limited understanding of lipid metabolism. The clarity of the manuscript will be greatly improved if the authors highlight/summarize prior publications that determined how HFD and corticosterone affect CPT1 enzyme levels and/or enzyme activity, acyl-coa levels, the rate of β -oxidation and glucose metabolism both in peripheral and CNS tissues. Again this will the reader a better understanding of where CPT1 target engagement needs to occur.

The presentation of the cylinder test and Y- Maze data in Figures 3 and 4 should match the format used in Figure 1 and 2 (inter-weaving of each group for each time point) to make the graphs easier to read and interpret. It should not be surprising that HFD effected cognition in ALS and WT mice (line 227) given that this has been studied extensively in WT mice.

The images presented in Figure 6 make it difficult to make any conclusions. Inserting arrows on current images and adding higher magnification images (20X) to point out differences between groups would be very helpful in helping the reader interpret the data. Semi-quantitative assessment of the staining would also be helpful (but is not required).

Is Etomoxir's MOA known? How do you explain that it reduces CPT1A staining in the spinal cord? Please discuss.

I found inclusion of the microbiome data confusing – I admit though I have limited knowledge of this field. I would consider removing the microbiome data from the current manuscript unless the authors believe that CPT1 targeting (inhibition) in the gut in combination with peripheral tissues and the CNS tissues contributes to the slowing disease progression in SOD1G93A mice. This is not discussed anywhere in the manuscript.

Why wasn't the WT Day 130 microbiome data included? This is needed to interpret Figures 7C-f. Was the effect of HFD and corticosterone on the microbiome evaluated? Did the authors considering what effect modulating the microbiome directly (e.g., via fecal transplant) to see what effect this has on CPT1 expression/activity in peripheral and CNS tissues?

Reviewer #2:

Remarks to the Author:

This manuscript describes new results showing the pathobiological roles of a lipid metabolism enzyme carnitine palmitoyl transferase 1 (CPT1) in a mouse of familial ALS (SOD1 G93A transgenic). The study introduces possible disease mechanisms driven by a new molecule, which are quite valuable for other researchers in the field. This reviewer accepts the significance and innovation of the data, but several major concerns still remain in the current manuscript. Particularly, experimental design and data presentation should be improved.

1. Most of animal studies did not have survival data. As ALS is a terminal disease, the lifespan result is commonly important to see the efficacy of specific treatments or experimental conditions in SOD1 G93A mice.

2. It was an interesting addition to describe the influence of CPT1 modification to the gut microbiota. However, this part of study is not well integrated (conceptionally and biologically) with the other animal results. And the authors do not address or discuss what specific mechanism are involved in the change of gut microbiota in SOD G93A mice. Furthermore, it is not clear whether the change of CPT1 or lipid metabolism primarily influenced to the population of gut microbiota, or whether the microbiota result was simply coming from the subsequent event associated with the changes of pathological conditions in the model.

3. Relevant to the comment above, the Discussion section is relatively abbreviated and does not cover the observations properly. Particularly, the paragraph about the microbiota is very short.

4. The study was done with female mice. A number of previous observations indicate that SOD1G93A rodent models show sexual dimorphism in disease progression and survival. It is equally worth to have males in the evaluation and see how sex influences the results.

5. The effect of corticosterone would be highly relied on its dose and treatment period. The author should clarify how "high" is the administered dose of corticosterone in terms of biological meaning. In other words, did the administration really simulate the blood level of corticosterone under high stress condition?

6. Some experimental groups were only having the limited number of animals (n=3 or 4).

7. Overall experimental methods are quite limited. Additional studies such as motor neuron survival, neuromuscular innervation, skeletal muscle pathology would further support the conclusion.

8. The immunohistological results in Figure 6 showed the change of GFAP-positive astrocytes. On the other hand, how was microglial reaction changed in these preparations? Based on the recent publication from the authors' laboratory (PLoS ONE, Jun 10; 15(6): e0234493, 2020), the inhibition of CPT1 may alter the level of inflammation. In ALS, neuroinflammation is one of pathological hallmarks during disease progression.

9. The manuscript needs significant improvements of data presentation.

- Commonly individual graphs are small in one figure.

- Some graphs have inconsistency in the order of the labeling. For instance, in Fig 1d-j SOD1 groups are coming first and WT ones are next. But the order is opposite in Figure 4e-i.

- Fig. 5: This figure is very busy and needs to be summarized for better clarification. Using a heat map, any table, or only using particular graphs in the main figure may be helpful.

- Fig. 6: Additional images with higher magnification will be much helpful to show individual cellular morphology and the change of protein expression. And there is no scale bar in the image.

- Some of figure legends look like a similar template with a simple list. They are not helpful for readers.

10. Title should include a word ALS, instead of the specific name of transgenic mouse model (SOD1 G93A mouse model).

11. Having 114 references is too much for a research article. Please re-check the list and only keep critical ones there.

Response to reviewers:

Dear Editor and reviewers, we thank you for your thorough review, valuable comments and insightful suggestions for our manuscript. We have taken those for the revisions and have prepared a point-by-point response here. All revisions are marked with **yellow** in the related manuscript file. We have prepared new figures for the whole manuscript and therefore these are not included in this file but in the manuscript file.

Response to reviewers' comments:

Reviewer #1 (Remarks to the Author):

Trabjerg et al. report that modulating carnitine palmitoyl transferase 1 (CPT1) activity effects disease progression in SOD1G93A mice, a familial mouse model of amyotrophic lateral sclerosis (ALS). CPT1 is an enzyme that facilitates the transport of medium-long chained fatty acids across the outer mitochondrial membrane to enable β -oxidation. There are 3 different isoforms of CPT1, each with varying levels of expression in different tissue types. CPT1A is expressed ubiquitously, CPT1B is limited to muscle and adipose tissue whereas CPT1C is found in the CNS. The authors show that either pharmacological inhibition of CPT1 activity with Etomoxir or genetic knockdown of CPT1A ameliorates disease symptoms in SOD1G93A mice. Additional experiments show that disease progression is accelerated in SOD1G93A mice following consumption of a 60% high fat diet or treatment with corticosterone. Finally, the authors show that downregulating CPT1 shifts the gut microbiota communities in SOD1G93A mice. Collectively, these findings suggest that CPT1 mediated lipid metabolism plays an important role in regulating disease progression in SOD1G93A mice and suggests modulating CPT1 might be a therapeutic target for ALS. Overall, I believe the results of these studies are valid, provide novel insights into the pathogenesis of ALS, and are of interest to scientists from different disciplines. However, methodological details of the experiments need to be addressed prior to being considered for publication (see below). The appropriate statistical tests were used for the pharmacological, genetic, HFD and corticosterone data. However, I am not in position to comment if the correct analyses were used for the microbiome data. Previous literature was cited correctly. Furthermore, the clarity and flow of the manuscript needs to be significantly improved especially for readers that are less familiar with lipid metabolism. Please see suggestions below.

Concerns

1. The title is somewhat subjective – get rid of strongly. Based on the results I would recommend changing the title to “Downregulating carnitine palmitoyl transferase 1 slows disease progression in the SOD1 G93A mouse model”.

Response: We have decided to change the title to “Downregulating carnitine palmitoyl transferase 1 affects disease progression in the SOD1 G93A mouse model of ALS” based on reviewer 1 and reviewer 2 (comment 10) suggestions.

2. The methods used to calculate the Neuroscore and survival needs to be explicitly stated in the methods. Interpretation of data presented in figures 1c, 2c, 2g, 3c and 4c is not possible without these details. I did not have access to the papers published referenced for these methods (which may also be true for potential readership).

Response: We thank the reviewer for noticing that other researchers might not have access to the referenced papers. We have explicitly stated the methods to calculate the neuroscore and survival in the method section, as follows:

” Mice were given a neurological score between zero to five as previously described^{3,4}. Zero = no tremor in hindlimbs and full extension of hindlimbs when suspended by its tail. One = tremor in hindlimbs and full extension of hindlimbs when suspended by its tail. Two = tremor in hindlimbs and unable to extend hindlimbs when suspended by its tail. Three = tremor in hindlimbs, unable to extend hindlimbs when suspended by its tail and wobbling gait. Four = tremor in hindlimbs, unable to extend hindlimbs when suspended by its tail difficulty walking with paralysis of one of both hindlimbs. Five = tremor in hindlimbs, unable to extend hindlimbs and unable to get up within 30s when placed on its side. Onset of disease was defined as the time point when tremor in the hind legs was present as previously described⁴. Due to ethical reasons and based on the guidelines in the Animal Facility, mice were terminated if they reached a neurological score of four or latest at day 160. Based on this, survival was defined as a neurological score below 4 at the final day of experimentation”

3. For Figures 1b, 2b, 3b and 4b the median age of tremor onset for each group needs to be reported along with the p values.

Response: We agree with the reviewer and have included median age of tremor onset for each group in the main text.

4. The survival data present in Figure 2G is incomplete. At what age were 0% of the genetically manipulated mice alive? The median survival ages for each group also need to be stated in the results.

Response: Due to ethical reasons, approval from the Danish Animal Inspectorate and based on the guidelines in the Animal Facility, mice were terminated if they reached a neurological score of four or latest at day 160. Based on this, survival was defined as a neurological score below 4 at the final day of experimentation. Due to the fact that mice had to be terminated at day 160, some mice did not reach a neurological score of 4 and therefore some of the groups have censored (“incomplete”) data in the survival analysis.

5. In the figure legends N values are given, but then it is stated the data represents 2-3 different animal studies (depending on the study). It is unclear if the N value represents the total number of mice used across all of the experiments or if it represents the N value per experiment. For example, for Figure #1, is the total N value 4-10 per group or 12-30 per group? Please clarify.

Response: We have clarified how the studies were repeated to evaluate reproducibility in the method section under "Statistics and reproducibility". The given N value *now* represents the total number of mice used across all of the experiments.

6. The target enzymes (CPT1A, B and C) and tissues (peripheral vs. CNS) of Etomoxir are unclear. Please edit the manuscript to include which isoforms of CPT1 are inhibited by Etomoxir and if it is known if the compound crosses the blood brain barrier. Similarly, for the genetic studies, the promoter used to manipulate CPT1A levels should be stated. We need to know if CPT1A expression and enzyme activity is altered in both peripheral and CNS tissues.

Response: Regarding etomoxir: we have now included a sentence stating:

" Etomoxir targets both CPT1A and CPT1B by antagonistic mechanism in the periphery and CNS and is indicated to cross the BBB based on functional analysis and chemical structure."

Regarding genetic studies: The mutation was made in the genomic *Cpt1a* gene and not based on promoter-*Cpt1a*-pA construct randomly integrated. In any case it is the *Cpt1a* promoter completely in its genomic context. So no expression level differences are expected in the different tissues. With respect to the *enzyme activity*: The mutation reduces the CPT1A activity to about 22%. So the activity of the wt would be 100%, the homozygous mutant ~22% and of the heterozygous wt/mut around 70%.

This has been clarified in the manuscript: "...we tested the effect of crossing *Cpt1a*^{P479L/P479L} mutated female mice with *SOD1* G93A male mice. The *Cpt1a*^{P479L/P479L} mouse strain was generated by mutating the genomic *Cpt1a* gene (genomic *Cpt1a* promoter), causing no expression changes in different tissues⁵. The mutation results in a CPT1A activity of approximately 22% compared to the wildtype protein in its' homozygote form and approximately 70% activity in the heterozygote form⁶"

7. The relative importance of CPT1 target engagement in peripheral vs CNS tissues should be included in the discussion. Making CNS penetrant compounds is difficult and if your data suggests this may not be necessary, this would be important to know for developing CPT1 inhibitors as potential therapeutics.

Response: We thank the reviewer for the comment and agree that this is essential to discuss. We have included the following section in the discussion: "A major problem in the treatment of neurodegenerative diseases is the development of drugs that can cross the BBB. Etomoxir is indicated to affect the metabolism both within the CNS⁷ and in the periphery⁸. Data

presented in the etomoxir experiments indicate that target engagement takes place both within the CNS and periphery regarding e.g. inflammation. However, another drug that downregulates β -oxidation is ranolazine, which has shown beneficial effects by downregulating CPT1B in the muscles in SOD1 G93A mice⁹ and inflammation *in vitro*¹⁰. Therefore, we *hypothesize* that peripheral target engagement in SOD1 G93A mice could prove to have effects by amelioration of peripheral hypermetabolism and thereby e.g. inflammation and oxidative stress.”

7. Oakes, N. D. *et al.* Development and initial evaluation of a novel method for assessing tissue-specific plasma free fatty acid utilization in vivo using (R)-2- bromopalmitate tracer. *J. Lipid Res.* **40**, 1155–1169 (1999).
 8. Timmers, S. *et al.* Augmenting muscle diacylglycerol and triacylglycerol content by blocking fatty acid oxidation does not impede insulin sensitivity. *Proc. Natl. Acad. Sci. U. S. A.* **109**, 11711–11716 (2012).
 9. Scaricamazza, S. *et al.* Skeletal-Muscle Metabolic Reprogramming in ALS-SOD1G93A Mice Predates Disease Onset and Is A Promising Therapeutic Target. *iScience* **23**, (2020).
 10. Aldasoro, M. *et al.* Effects of ranolazine on astrocytes and neurons in primary culture. *PLoS One* **11**, 1–15 (2016).
8. Experiments 3 and 4 complement the first 2 studies nicely and are in agreement with some of the previously published work in the same model. Presumably, experiments 3 (HFD study) and 4 (corticosterone study) were carried out to show that promoting acyl-coa accumulation, increasing the rate of β -oxidation, and impairing glucose metabolism have the opposite effect of down-regulating CPT1 enzyme activity (experiments 1 and 2). This connection between experiments may be challenging for individuals that have a more limited understanding of lipid metabolism. The clarity of the manuscript will be greatly improved if the authors highlight/summarize prior publications that determined how HFD and corticosterone affect CPT1 enzyme levels and/or enzyme activity, acyl-coa levels, the rate of β -oxidation and glucose metabolism both in peripheral and CNS tissues. Again this will the reader a better understanding of where CPT1 target engagement needs to occur.

Response: We thank the reviewer for this valuable comment. We agree that the connection between experiments may be challenging for individuals that have a more limited understanding of lipid metabolism. Therefore, we have included the following paragraph in the new section 2.6:

” Downregulation of CPT1 isoforms A and B activity by etomoxir (**Figure 1, 2**) and CPT1A activity by the *Cpt1a* P479L mutation (**Figure 5-6**) delayed disease progression and attenuated multiple pathological mechanisms including inflammation and oxidative stress in the SOD1 G93A mouse model. Based on this we hypothesized that the upregulation of CPT1A and CPT1B activity would exacerbate disease progression and severity. High fat diets (HFD) with a high level of saturated fatty acids increases the level of acyl-CoA¹¹, CPT1A¹² and CPT1B activity¹³ and thereby β -oxidation in the periphery including liver and muscle tissue. In addition, HFD increases pyruvate dehydrogenase kinase-4 activity and thereby

*downregulates the pyruvate dehydrogenase complex and thus the utilization of pyruvate in the glucose oxidation process*¹⁴. *HFD also downregulates the transport of glucose into the CNS and increases insulin resistance in the periphery and CNS, which results in decreased glucose metabolism*¹⁵. ”

11. Liu, X. *et al.* High-resolution metabolomics with acyl-CoA profiling reveals widespread remodeling in response to diet. *Mol. Cell. Proteomics* **14**, 1489–1500 (2015).
12. Kakimoto, P. A. & Kowaltowski, A. J. Effects of high fat diets on rodent liver bioenergetics and oxidative imbalance. *Redox Biol.* **8**, 216–225 (2016).
13. Yun, H. Y., Lee, T. & Jeong, Y. High-Fat Diet Increases Fat Oxidation and Promotes Skeletal Muscle Fatty Acid Transporter Expression in Exercise-Trained Mice. *J. Med. Food* **23**, 281–288 (2020).
14. Rinnankoski-Tuikka, R. *et al.* Effects of high-fat diet and physical activity on pyruvate dehydrogenase kinase-4 in mouse skeletal muscle. *Nutr. Metab.* **9**, 1–13 (2012).
15. Liu, Z. *et al.* High-fat diet induces hepatic insulin resistance and impairment of synaptic plasticity. *PLoS One* **10**, 1–16 (2015).

In the new paragraph 2.8 considering the effects of corticosterone we have added the following sentences:

” *Any imbalances to an organism’s homeostasis elicit a complex stress response that causes activation of the neuroendocrine and autonomic system. One of the essential systems in the stress response is the hypothalamic-pituitary-adrenal (HPA) axis*¹⁶. *During acute stress, such as critical sickness, the stress response is beneficial for survival. However, prolonged stress, due to psychological or physiological circumstances, causes an overactivation of the HPA-axis resulting in chronic high levels of glucocorticoids (cortisol). High levels of cortisol, in turn, result in increased activity of PDK4, which downregulates glucose metabolism*¹⁷. *In addition, cortisol promotes insulin resistance, which forces metabolism towards lipolysis*^{18,19} *and increased β -oxidation*²⁰. *These findings could indicate that increased levels of glucocorticoids could increase the activity of CPT1. Several studies support the importance of a dysregulation of the HPA-axis in ALS and the SOD1 G93A mouse model. ALS patients have been found to have increased morning cortisol blood levels and a disruption of the cortisol awakening response*^{21,22}. *Studies have shown that prolonged high levels of cortisol leads to glucocorticoid receptor resistance which eventually results in failure to down-regulate the inflammatory response*²³. *Chronic restraint stress in SOD1 G93A mice increased the level of corticosterone, which correlated with decreased survival and increased inflammation*²⁴.

Based on this, we hypothesized that increased levels of glucocorticoids would result in a downregulation of glucose metabolism and an upregulation of CPT1-mediated lipid metabolism. Therefore, we tested the effect of administering the mouse equivalent of cortisol, corticosterone, to SOD1 G93A mice from day 70 and until day 100 (Figure 9a). ”

16. Bellavance, M. A. & Rivest, S. The HPA - immune axis and the immunomodulatory actions

- of glucocorticoids in the brain. *Front. Immunol.* **5**, 1–13 (2014).
17. Kuo, T., Harris, C. A. & Wang, J. C. Metabolic functions of glucocorticoid receptor in skeletal muscle. *Mol. Cell. Endocrinol.* **380**, 79–88 (2013).
 18. Djurhuus, C. B. *et al.* Effects of cortisol on lipolysis and regional interstitial glycerol levels in humans. *Am. J. Physiol. - Endocrinol. Metab.* **283**, E172–E177 (2002).
 19. Lönnqvist, F., Wennlund, A., Wahrenberg, H. & Arner, P. Effects of Mental Stress on Lipolysis in Humans. *Metabolism* **41**, 622–630 (1992).
 20. Divertie, G. D., Jensen, M. D. & Miles, J. M. Stimulation of lipolysis in humans by physiological hypercortisolemia. *Diabetes* **40**, 1228–32 (1991).
 21. Spataro, R. *et al.* Plasma cortisol level in amyotrophic lateral sclerosis. *J. Neurol. Sci.* **358**, 282–286 (2015).
 22. Roozendaal, B. *et al.* The cortisol awakening response in amyotrophic lateral sclerosis is blunted and correlates with clinical status and depressive mood. *Psychoneuroendocrinology* **37**, 20–26 (2012).
 23. Cohen, S. *et al.* Chronic stress, glucocorticoid receptor resistance, inflammation, and disease risk. *Proc. Natl. Acad. Sci.* **109**, 5995–5999 (2012).
 24. Fidler, J. A. *et al.* Disease progression in a mouse model of amyotrophic lateral sclerosis: the influence of chronic stress and corticosterone. *FASEB J.* **25**, 4369–4377 (2011).
9. The presentation of the cylinder test and Y-maze data in Figures 3 and 4 should match the format used in Figure 1 and 2 (inter-weaving of each group for each time point) to make the graphs easier to read and interpret. It should not be surprising that HFD effected cognition in ALS and WT mice (**line 227**) given that this has been studied extensively in WT mice.

Response: We have changed the presentation of the cylinder test and y-maze data so it matches the format used in the old figure 1 and figure 2 (new figure 1 and 3). In addition, we have changed the wording of the sentence regarding HFD and cognition to:

” Both *Wt+HFD* and *SOD1+HFD* had a significant decrease in their visuospatial memory based on the y-maze test, which is in agreement with previous studies in *Wt* mice receiving HFD”.

10. The images presented in Figure 6 make it difficult to make any conclusions. Inserting arrows on current images and adding higher magnification images (20X) to point out differences between groups would be very helpful in helping the reader interpret the data. Semi-quantitative assessment of the staining would also be helpful (but is not required).

Response: We have included new images with 10X magnification and added arrows and asterisks on the old 2.5X images and new images. The 2.5X images are now placed in the supplementary figures.

11. Is Etomoxir's MOA known? How do you explain that it reduces CPT1A staining in the spinal cord? Please discuss.

Response: We agree that the MOA of etomoxir needs to be discussed. We have included these aspects in multiple sections of the discussion.

12. I found inclusion of the microbiome data confusing – I admit though I have limited knowledge of this field. I would consider removing the microbiome data from the current manuscript unless the authors believe that CPT1 targeting (inhibition) in the gut in combination with peripheral tissues and the CNS tissues contributes to the slowing disease progression in SOD1G93A mice. This is not discussed anywhere in the manuscript.

Response: We believe that inhibition of CPT1 affects the gut microbiome directly but also indirectly by the changes in metabolites in e.g. serum and that this affects disease mechanisms in the periphery and CNS. It has previously been shown that the gut microbiome in the SOD1 G93A mouse model is correlated to disease mechanisms such as inflammation, motor neuron and muscle pathogenesis²⁵. We have included these aspects in the microbiome result section and discussion.

21. Figueroa-Romero, C. *et al.* Temporal evolution of the microbiome, immune system and epigenome with disease progression in ALS mice. *DMM Dis. Model. Mech.* **13**, (2020).

13. Why wasn't the WT Day 130 microbiome data included? This is needed to interpret Figures 7C-f. Was the effect of HFD and corticosterone on the microbiome evaluated? Did the authors considering what effect modulating the microbiome directly (e.g., via fecal transplant) to see what effect this has on CPT1 expression/activity in peripheral and CNS tissues?

Response: We have included the Wt day 130 microbiome data and conducted 16s rRNA sequencing on fecal samples from the HFD and corticosterone studies and included these data in the manuscript. We have not conducted fecal transplant but we agree that this will be highly relevant to perform in future studies.

Reviewer #2 (Remarks to the Author):

This manuscript describes new results showing the pathobiological roles of a lipid metabolism enzyme carnitine palmitoyl transferase 1 (CPT1) in a mouse of familial ALS (SOD1 G93A transgenic). The study introduces possible disease mechanisms driven by a new molecule, which are quite valuable for other researchers in the field. This reviewer accepts the significance and innovation of the data, but several major concerns still remain in the current manuscript. Particularly, experimental design and data presentation should be improved.

1. Most of animal studies did not have survival data. As ALS is a terminal disease, the lifespan result is commonly important to see the efficacy of specific treatments or experimental conditions in SOD1 G93A mice.

Response: We agree with the reviewer and have included survival data for etomoxir, *Cpt1a* P479L and HFD studies in the figures and included significance levels and median survival in the main text. The corticosterone study was terminated before survival, as we were interested in the effect on disease onset and progression.

2. It was an interesting addition to describe the influence of CPT1 modification to the gut microbiota. However, this part of study is not well integrated (conceptionally and biologically) with the other animal results. And the authors do not address or discuss what specific mechanism are involved in the change of gut microbiota in SOD G93A mice. Furthermore, it is not clear whether the change of CPT1 or lipid metabolism primarily influenced to the population of gut microbiota, or whether the microbiota result was simply coming from the subsequent event associated with the changes of pathological conditions in the model.

Response: We thank the reviewer for the comment. We have tried to improve the integration of the relevance of the microbiome section with the other animal results. We believe that inhibition of CPT1 affects the gut microbiome directly but also indirectly by changes in metabolites in e.g. serum and that this affects disease mechanisms in the periphery and within the CNS. We have published data that in *Cpt1a* P479L mutant animals, even though they are backcrossed several times to C57Bl/6J background, due to a downregulated lipid metabolism there is a shift in gut microbiota, showing that metabolites directly influence gut microbiota²⁶. It has previously been shown that the gut microbiome in the SOD1 G93A mouse model is correlated to disease mechanisms such as inflammation, motor neuron and muscle pathogenesis²⁵. We have included these aspects in the microbiome result section and discussion.

25. Figueroa-Romero, C. *et al.* Temporal evolution of the microbiome, immune system and epigenome with disease progression in ALS mice. *DMM Dis. Model. Mech.* **13**, (2020).
26. Trabjerg, M. S. *et al.* Dysregulation of metabolic pathways by carnitine palmitoyl-transferase 1 plays a key role in central nervous system disorders: experimental evidence based on animal models. *Sci Rep* **10**, 1–19 (2020).

3. Relevant to the comment above, the Discussion section is relatively abbreviated and does not cover the observations properly. Particularly, the paragraph about the microbiota is very short.

Response: We agree with the reviewer. We have restructured the discussion and included new paragraphs considering the mode of action of etomoxir, target engagement and relevance of the microbiota.

4. The study was done with female mice. A number of previous observations indicate that SOD1G93A rodent models show sexual dimorphism in disease progression and survival. It is equally worth to have males in the evaluation and see how sex influences the results.

Response: We have included males in the analysis of survival for SOD1 G93A males compared to SOD1Cpt1a/Cpt1a mutant males. We acknowledge the valid comments of the editor and reviewers regarding sex, however, sexual dimorphism does not affect disease progression and survival in the B6.Cg-Tg(SOD1*G93A)1Gur/J mouse model in contrast to the B6SJL-Tg(SOD1*G93A)1Gur/J as described by a meta-analysis ¹ and a large *in vivo* study ².

We have included this sentence in the result section and data (Supplementary Figure 2) illustrating no difference in our SOD1 G93A cohort:

” Sexual dimorphism have been reported to oppose a problem in the B6SJL-Tg(SOD1*G93A)1Gur/J mouse model but not in the B6.Cg-Tg(SOD1*G93A)1Gur/J model ¹, which was used in this study. Accordingly, we did not find any difference in clinical-relevant phenotype between male and female SOD1 mice at day 140 nor survival (**Figure S2**).”

1. Pfohl, S. R., Halicek, M. T. & Mitchell, C. S. Characterization of the Contribution of Genetic Background and Gender to Disease Progression in the SOD1 G93A Mouse Model of Amyotrophic Lateral Sclerosis: A Meta-Analysis. *J. Neuromuscul. Dis.* **2**, 137–150 (2015).
2. Heiman-Patterson, T. D. *et al.* Background and gender effects on survival in the TgN(SOD1-G93A)1Gur mouse model of ALS. *J. Neurol. Sci.* **236**, 1–7 (2005).

5. The effect of corticosterone would be highly relied on its dose and treatment period. The author should clarify how “high” is the administered dose of corticosterone in terms of biological meaning. In other words, did the administration really simulate the blood level of corticosterone under high stress condition?

Response: We have measured the level of corticosterone in serum at day 100 and found results in accordance with previous studies assessing the role of hyperactivated HPA-axis and corticosterone in the SOD1 G93A mouse model (21).

21. Fidler, J. A. *et al.* Disease progression in a mouse model of amyotrophic lateral sclerosis: the influence of chronic stress and corticosterone. *FASEB J.* **25**, 4369–4377 (2011).

6. Some experimental groups were only having the limited number of animals (n=3 or 4).

Response: We have increased sample size where possible. However, due to the nature of the experiments, logistics, the SOD1 G93A model, ethical approval and costs is it not possible to increase sample size in all experiments.

7. Overall experimental methods are quite limited. Additional studies such as motor neuron survival, neuromuscular innervation, skeletal muscle pathology would further support the conclusion.

Response: We have conducted the following extra experiments to address the concerns raised by the Editor and reviewers:

- Microglial reaction: We have measured CX3CR1, IL-10, IL-1 β and TNF- α cytokine levels in the spinal cord. In addition, we have performed IHC for Iba1 labeling in the spinal cord of SOD1 G93A mice treated with etomoxir and placebo. Moreover, we have performed RT-qPCR to evaluate the fold change expression of CD68 in the corticosterone study to evaluate the level of reactive microglia.
 - Motor neuron survival: We have measured the serum level of neurofilament light-chain as this is a biomarker of neurodegenerative diseases including ALS, marker of disease progression and death of large neurons with myelinated axons (including motor neurons). In addition, we have evaluated the protein level of Choline O-acetyltransferase (ChAT) in the spinal cord of the SOD1 G93A etomoxir experiment, *Cpt1a* P479L studies and SOD1 high fat diet studies. Additionally, we have performed immunohistochemistry for ChAT labelling in the spinal cord of SOD1 G93A mice treated with etomoxir and placebo.
 - Neuromuscular innervation, and skeletal muscle pathology: We have evaluated muscle atrophy and denervation based on normalized weight of tibialis anterior muscles, measured MuSK as a marker of denervation together with gene expression of *Chran1a* and measured MuRF1 as a marker of muscle atrophy. We did not perform IHC on muscle tissue and therefore we cannot provide histological analysis of muscle tissue. However, we have measured IL-10, IL-1 β and TNF- α cytokine levels in tibialis anterior muscles in multiple of the experiments to evaluate inflammatory effects of downregulating CPT1A/B activity both in the central nervous system and in the periphery.
8. The immunohistological results in Figure 6 showed the change of GFAP-positive astrocytes. On the other hand, how was microglial reaction changed in these preparations? Based on the recent publication from the authors' laboratory (PLoS ONE, Jun 10; 15(6): e0234493, 2020), the inhibition of CPT1 may alter the level of inflammation. In ALS, neuroinflammation is one of pathological hallmarks during disease progression.

Response: We have included new images with 10X magnification and added arrows and asterisks on the old 2.5X images (now placed in "Supplementary figures") and new images. The 2.5X images are now placed in the supplementary figures. In addition, we have performed immunohistological staining for IBA1 and CHAT on spinal cord sections (new figure 4).

9. The manuscript needs significant improvements of data presentation.

- Commonly individual graphs are small in one figure.

Response: We have increased the size of individual graphs wherever possible.

- Some graphs have inconsistency in the order of the labeling. For instance, in Fig 1d-j SOD1 groups are coming first and WT ones are next. But the order is opposite in Figure 4e-i.

Response: We have corrected the labeling to assure consistency.

- Fig. 5: This figure is very busy and needs to be summarized for better clarification. Using a heat map, any table, or only using particular graphs in the main figure may be helpful.

Response: We have divided the figure into sub-figures and reformatted it to heatmaps for clarity (new figure 2, 6, 8).

- Fig. 6: Additional images with higher magnification will be much helpful to show individual cellular morphology and the change of protein expression. And there is no scale bar in the image.

Response: We have included new images with 10X magnification and added arrows, asterisks and scale bar on the old 2.5X images and new images. The 2.5X images are now placed in the supplementary figures.

- Some of figure legends look like a similar template with a simple list. They are not helpful for readers.

Response: We have evaluated all figure legends and restructured/rewritten them.

10. Title should include a word ALS, instead of the specific name of transgenic mouse model (SOD1 G93A mouse model).

Response: We have decided to change the title to “Downregulating carnitine palmitoyl transferase 1 affects disease progression in the SOD1 G93A mouse model of ALS” based on reviewer 1 (comment 1) and reviewer 2 (comment 10) suggestions

11. Having 114 references is too much for a research article. Please re-check the list and only keep critical ones there.

Response: We have re-checked all the references and only kept critical ones.

Reviewers' Comments:

Reviewer #1:

Remarks to the Author:

The revised manuscript is significantly improved and suitable for publication.

Reviewer #2:

Remarks to the Author:

This reviewer appreciate the authors to properly respond to most of the listed concerns. While the revised manuscript significantly improves a number of points, there are still a few issues specifically for the presentation and quality of the histology results.

1. Commonly, the new 10x images are still small without high-power images. The current images cannot clearly see individual cellular morphology. The arrows and asterisks in images are not helpful enough. Specifically, it seems that some arrows in ChAT staining are pointing outside of the ventral horns where motor neurons should be located. Furthermore, the asterisks in the same images are not nicely indicating motor neurons. The presentation of these figures should be improved.

2. Neuromuscular innervation and skeletal muscle pathology are still evaluated only by the expression of limited genes. Without immunohistochemistry, it is not uncertain whether gene expression really represent the level of motor endplate innervation and muscle atrophy.

Response to reviewers:

Dear Editor and reviewers, we thank you for your thorough review, valuable comments and insightful suggestions for our manuscript. We have taken those for the revisions and have prepared a point-by-point response here. All revisions are marked with **yellow** in the related manuscript file. We have prepared new figures (2, 3, 4 and 6) and have included these in this file and in the revised manuscript file.

Response to reviewers' comments:

Reviewer #1 (Remarks to the Author):

The revised manuscript is significantly improved and suitable for publication.

Response: We thank the reviewer for the acknowledgement of the revisions and for the spending time to review our work.

Reviewer #2 (Remarks to the Author):

This reviewer appreciate the authors to properly respond to most of the listed concerns. While the revised manuscript significantly improves a number of points, there are still a few issues specifically for the presentation and quality of the histology results.

Comment 1. Commonly, the new 10x images are still small without high-power images. The current images cannot clearly see individual cellular morphology. The arrows and asterisks in images are not helpful enough. Specifically, it seems that some arrows in ChAT staining are pointing outside of the ventral horns where motor neurons should be located. Furthermore, the asterisks in the same images are not nicely indicating motor neurons. The presentation of these figures should be improved.

Response: We agree with the reviewer that the 10x images was too small. Therefore, we have now provided x16 high-power magnification images, which illustrates immunohistochemistry differences between the groups. These pictures are placed below: “

Downregulation of CPT1 lipid metabolism modulates CPT1A and CPT1C labeling, myelination, astrogliosis, reactive microglia and motor neuron survival in the spinal cord in SOD1 G93A mice

Following the clinical-relevant effects of downregulating CPT1 lipid metabolism by etomoxir in the SOD1 G93A mouse model (**Figure 1**) and the changes in metabolic, inflammatory and oxidative stress markers in the lumbar spinal cord (**Figure 2**), we performed an immunohistochemical peroxidase staining on the lumbar spinal cord to evaluate morphological changes. To evaluate the effects of CPT1 downregulation, SOD1 G93A mice received etomoxir (SOD1+E) or placebo (SOD1+P) from day 70 and until termination at day 130. SOD1+E mice had significantly lower neuroscore and increased grip strength compared to SOD1+P mice (**Figure 11-m**). First, we evaluated whether etomoxir affected the labeling of CPT1A. We found that SOD1+P mice had

increased labeling compared to SOD1+E and Wt (**Figure 3a-c, Figure S4**), which indicated that etomoxir decreased the level of CPT1A, as previously described²³. Next, we evaluated whether etomoxir affected the labeling of CPT1C. **However, we did not find any indications of difference in the labeling (Figure 3d-e, Figure S4)**. Previously, we and others have shown that CPT1 downregulation results in diminished demyelination^{13,20}. Further, demyelination has been observed in the SOD1 G93A mouse model- and ALS patients^{50,51}. Therefore, we performed staining for MBP. In agreement with the gene expression studies, SOD1+P mice were found to have decreased labeling of MBP, which was counteracted by etomoxir (**Figure 3g-i, Figure S4**).

Figure 3: Immunohistochemical staining in the lumbar spinal from the SOD1 G93A etomoxir experiment

(a-c) CPT1A staining in lumbar spinal cord from Wt, SOD1+P and SOD1+E mice at day 130 indicating increased labeling in SOD1+P mice (arrows) and pathological morphology of neurons (asterisks). **(d-f)** CPT1C staining in lumbar spinal cord from Wt, SOD1+P and SOD1+E mice at day 130 indicating no difference in the labeling in SOD1+P mice but differences in the morphology of neurons (asterisks). **(g-i)** MBP staining in lumbar spinal cord from Wt, SOD1+P and SOD1+E mice at day 130 indicating decreased labeling in SOD1+P mice (arrows). All images are presented with 16x magnification. N = 2 – 4 animals per group. WT=wild-type, SOD1 = SOD1 G93A genotype, E=etomoxir, P=placebo.

Spinal cords from ALS patients and SOD1 mouse model are characterized by astrogliosis and increased reactive astrocytes⁵². Therefore, we assessed the labeling of the reactive astrocyte marker, GFAP, in the spinal cord of SOD1 mice. SOD1+P mice had increased labeling of GFAP compared to Wt and SOD1+E mice (**Figure 4a-c, Figure S4**), indicating increased reactive astrocytes in the SOD1+P group. Another hallmark of disease in ALS and the SOD1 G93A model is neuroinflammation¹. Therefore, we evaluated whether etomoxir altered the labeling of IBA1, a marker of reactive microglia and macrophages, in the lumbar spinal cord. SOD1+E mice had decreased labeling compared with SOD1+P mice (**Figure 4d-f**), which was in accordance with the inflammatory data presented in **Figure 2**. Finally, we assessed the labeling of ChAT, a marker of cholinergic neurons, in the ventral horn in the lumbar spinal cord and found indications of decreased labeling in SOD1+P compared to SOD1+E mice (**Figure 4g-i**). Overall, the immunohistochemical staining corresponded to the data presented in **Figure 1** and **Figure 2**.

Figure 4: Immunohistochemical staining in the lumbar spinal from the SOD1 G93A etomoxir experiment

(a-c) GFAP staining in lumbar spinal cord from Wt, SOD1+P and SOD1+E mice at day 130 indicating increased number of reactive astrocytes in the ventral horn in SOD1+P mice (arrows) compared to Wt and SOD1+E mice. **(d-f)** IBA1 staining in lumbar spinal cord from Wt, SOD1+P and SOD1+E mice at day 130 indicating increased labeling and infiltration of reactive microglia in the ventral horn in SOD1+P mice (arrows) compared to SOD1+E and Wt mice. **(g-i)** ChAT staining

in lumbar spinal cord from Wt, SOD1+P and SOD1+E mice at day 130 indicating decreased labeling in SOD1+P mice (arrows) and pathological morphology of neurons. All images are presented with 16x magnification. N = 2 – 4 animals per group. WT=wild-type, SOD1 = SOD1 G93A genotype, E=etomoxir, P=placebo.

”

Comment 2. Neuromuscular innervation and skeletal muscle pathology are still evaluated only by the expression of limited genes. Without immunohistochemistry, it is not uncertain whether gene expression really represent the level of motor endplate innervation and muscle atrophy.

Response: We agree with the reviewer that gene expression is not always the same as protein expression. Protein expression analysis to confirm the gene expression is preferable. This protein expression can be analyzed in immunohistochemistry or as we preferred ELISA analysis. During the embedding and sectioning of the tibialis anterior muscles the tissues were deemed not suitable for staining as the tissue shattered. Therefore, we decided to confirm the muscle pathology (atrophy) and denervation we measured by ELISA on tissue homogenate. We agreed this with the Editor. We have evaluated the protein levels of atrogen-1 (atrophy marker; DOI: [10.7554/eLife.10528](https://doi.org/10.7554/eLife.10528)) and myogenin (markers of denervation; DOI: [10.1016/j.isci.2020.101087](https://doi.org/10.1016/j.isci.2020.101087)) using tissue homogenate to provide some data to clarify the level of atrophy and denervation (New figure 2o-q and figure 6o-q). This experiments have been conducted for the etomoxir (Figure 1) and SOD1 x *Cpt1a* P479L (Figure 5) studies as we did not have available tissue to conduct these assays for the high fat diet experiments. We acknowledge that these new experiments do not give us the spatial information but still substantiates the behaviour and gene expression data. We have included the following paragraphs and figures in the manuscript:

“... In addition, we investigated differences in the protein level of muscle RING-finger protein-1 (MuRF-1) and atrogen-1 (**Figure 2o**), which both represents markers of atrophy. Wt mice had a significant lower level of atrogen-1 in accordance with the muscle atrophy evaluation in **Figure 2n** but no difference was observed between the treatment groups. Next, we evaluated the protein level of muscle skeletal receptor tyrosine-protein kinase (MuSK), which protects against denervation⁴⁹. SOD1+P mice had a significant decreased level compared to Wt mice this was not the case for SOD1+E mice (**Figure 2p**). Further, we evaluated the protein level of myogenin, which increases during denervation⁵⁰. The Wt mice had significantly lower levels compared to SOD1+P mice and the SOD1+E mice had significantly lower levels compared to SOD1+P mice (**Figure 2q**). This could indicate that downregulation of CPT1 lipid metabolism could play a role in diminishing denervation. However, this will require further investigations.

Figure 2: Downregulation of CPT1 activity by etomoxir potentially shifts metabolism towards glucose utilization and ameliorate disease mechanisms including inflammation, mitochondrial dysfunction and oxidative stress. **a)** Serum glucose levels. Mean mmol/L ± SEM. **b)** Serum LDL levels. Mean mmol/L ± SEM. **c)** Serum HDL levels. Mean mmol/L ± SEM. **d)** Serum LDL/HDL ratio levels. Mean mmol/L ratio ± SEM. **e)** Serum NF-L levels. Mean pg/mL ± SEM, **f)** ChAT levels in lumbar spinal cord tissue homogenate. Median ng/mg total protein ± IQR. **g)** CX3CR1 levels in lumbar spinal cord tissue homogenate. Mean ng/mg total protein ± SEM. **h)** IL-10 levels in lumbar spinal cord tissue homogenate. Mean pg/mg total protein ± SEM. **i)** IL-1β levels in lumbar

spinal cord tissue homogenate. Mean pg/mg total protein \pm SEM. **j)** TNF- α levels in lumbar spinal cord tissue homogenate. Mean pg/mg total protein \pm SEM. **k-m)** Fold change gene expression of metabolic, glial, inflammatory and oxidative stress genes in lumbar spinal cord tissue. Mean normalized fold change gene expression \pm SEM. **n)** Weight of tibialis anterior muscle at termination. Weight of tissue were normalized to body weight and expressed as mean \pm SEM. **o)** MuRF1 and atrogin-1 levels in tibialis anterior tissue homogenate. Mean ng/mg total protein \pm SEM. **p)** MuSK levels in tibialis anterior tissue homogenate. Mean ng/mg total protein \pm SEM. **q)** Myogenin levels in tibialis anterior tissue homogenate. Mean ng/mg total protein \pm SEM. **r)** IL-10 levels in tibialis anterior tissue homogenate. Mean pg/mg total protein \pm SEM. **s)** IL-1 β levels in tibialis anterior tissue homogenate. Mean pg/mg total protein \pm SEM. **t)** TNF- α levels in tibialis anterior tissue homogenate. Mean pg/mg total protein \pm SEM. **u)** Fold change gene expression of metabolic, inflammatory, oxidative stress and denervation genes in tibialis anterior tissue homogenate. Mean normalized fold change gene expression \pm SEM. Serum samples and tissue were obtained at termination. All data was analyzed using one-way ANOVA followed by Tukey post hoc test or Kruskal-Wallis test followed by Dunns post hoc test. N=5-8 for serum analysis and 3-5 for all other experiments. Data are representative of one experiment. Gene expression was normalized to *β -actin* and *Gapdh*. * Significant differences between groups in all analyses except gene expression experiments. * $p \leq 0.05$; ** $p \leq 0.01$. Significant annotations in gene expression experiments (one sign= $p \leq 0.05$, two signs= $p \leq 0.01$). *=SOD1+P vs SOD1+E, %=Wt+P vs SOD1+P, #=Wt+P vs SOD1+E. Wt=wildtype, SOD1 = SOD1 G93A genotype, E=etomoxir, P=placebo, SEM=standard error of mean, IQR=interquartile range, LDL=low-density lipoproteins, HDL=High-density lipoproteins, NF-L=Neurofilament light-chain, ChAT=Choline o-acetyltransferase, MuSK= Muscle skeletal receptor tyrosine-protein kinase, MuRF1= Muscle RING-finger protein-1.

... In addition, SOD1^{Wt/Cpt1a} and SOD1^{Cpt1a/Cpt1a} mice had lower levels of MuRF1 (Figure 6o) in accordance with the higher normalized weight of muscle tissue (Figure 6n). Further, SOD1^{Wt/Cpt1a} mice had a tendency towards a significantly lower level of the atrophy marker atrogin-1¹⁶ compared to SOD1 mice (Figure 6o). Next, we evaluated the protein level of MuSK and found that SOD1^{Cpt1a/Cpt1a} mice had higher levels compared to SOD1 mice (Figure 6p). In addition, SOD1^{Wt/Cpt1a} and SOD1^{Cpt1a/Cpt1a} mice had significantly lower protein levels of myogenin, which

could indicate a lower degree of denervation in these mice (**Figure 6q**). However, this will require further experiments to confirm.

Figure 6: Downregulation of CPT1A activity by *Cpt1a* P479L mutations potentially shifts metabolism towards glucose utilization and ameliorate disease mechanisms including inflammation, mitochondrial dysfunction and oxidative stress. **a)** Serum glucose levels. Mean mmol/L \pm SEM. **b)** Serum LDL levels. Mean mmol/L \pm SEM. **c)** Serum HDL levels. Mean mmol/L \pm SEM. **d)** Serum LDL/HDL ratio levels. Mean mmol/L ratio \pm SEM. **e)** Serum NF-L levels. Mean pg/mL \pm SEM. **f)** ChAT levels in lumbar spinal cord tissue homogenate. Mean ng/mg total protein \pm SEM. **g)** CX3CR1 levels in lumbar spinal cord tissue homogenate. Mean ng/mg total protein \pm SEM. **h)** IL-10 levels in lumbar spinal cord tissue homogenate. Median pg/mg total protein \pm IQR. **i)** IL-1 β levels in lumbar spinal cord tissue homogenate. Mean pg/mg total protein \pm SEM. **j)** TNF- α levels in lumbar spinal cord tissue homogenate. Mean pg/mg total protein \pm SEM. **k-m)** Fold change gene expression of metabolic, glial, inflammatory and oxidative stress genes in lumbar spinal cord tissue. Mean normalized fold change gene expression \pm SEM. **n)** Weight of tibialis anterior muscle at termination. Weight of tissue were normalized to body weight and expressed as mean \pm SEM. **o)** MuRF1 and atrogin-1 levels in tibialis anterior tissue homogenate. Mean ng/mg total protein \pm SEM. **p)** MuSK levels in tibialis anterior tissue homogenate. Mean ng/mg total protein \pm SEM. **q)** Myogenin levels in tibialis anterior tissue homogenate. Mean ng/mg total protein \pm SEM. **r)** IL-10 levels in tibialis anterior tissue homogenate. Mean pg/mg total protein \pm SEM. **s)** IL-1 β levels in tibialis anterior tissue homogenate. Mean pg/mg total protein \pm SEM. **t)** TNF- α levels in tibialis anterior tissue homogenate. Mean pg/mg total protein \pm SEM. **u)** Fold change gene expression of metabolic, inflammatory, oxidative stress and denervation genes in tibialis anterior tissue homogenate. Mean normalized fold change gene expression \pm SEM. Serum samples and tissue were obtained at termination. All data was analyzed using one-way ANOVA followed by Tukey post hoc test or Kruskal-Wallis test followed by Dunns post hoc test. N=3-8 for serum analysis and 3-5 for all other experiments. Data are representative of one experiment. Gene expression was normalized to *β -actin* and *Gapdh*. * Significant differences between groups in all analyses except gene expression experiments. * $p \leq 0.05$; ** $p \leq 0.01$. Significant annotations in gene expression experiments (one sign= $p \leq 0.05$, two signs= $p \leq 0.01$). *=SOD1 vs SOD1^{*Cpt1a/Cpt1a*}, %=SOD1^{*Wt/Cpt1a*} vs SOD1^{*Cpt1a/Cpt1a*}, #=SOD1 vs SOD1^{*Wt/Cpt1a*}. SOD1=SOD1 G93A genotype, SOD1^{*Wt/Cpt1a*} = SOD1 G93A mice with heterozygote *Cpt1a* P479L mutation, SOD1^{*Cpt1a/Cpt1a*} = SOD1 G93A mice with homozygote *Cpt1a* P479L mutation, SEM=standard error of mean, IQR=interquartile range, LDL=low-density lipoproteins, HDL=High-density lipoproteins, NF-L=Neurofilament light-chain, ChAT=Choline o-acetyltransferase, MuSK= Muscle skeletal receptor tyrosine-protein kinase, MuRF1= Muscle RING-finger protein-1.

”

Reviewers' Comments:

Reviewer #2:

None